# CorrSteer: Generation-Time LLM Steering via Correlated Sparse Autoencoder Features

**Seonglae Cho** [1 2]  **Zekun Wu** [1 2]  **Adriano Koshiyama** [1 2]

## Abstract

Sparse Autoencoders (SAEs) decompose LLM activations into interpretable features, yet existing SAE-based steering methods require contrastive datasets or large activation stores. We introduce CorrSteer, which selects steering features by correlating task outcomes with SAE activations computed during generation, then validates these selections through intervention. This two-stage approach treats correlation as a selection heuristic and intervention as the causal test: features that both correlate with success and improve performance when amplified are retained. Coefficients derive from mean activations on correct samples, yielding a fully automated pipeline without task-specific tuning. On Gemma-2 2B and LLaMA-3.1 8B, CorrSteer achieves +3.3% on MMLU (4k samples) and +27.1% on HarmBench (108 samples), with lower side-effect ratios than fine-tuning despite comparable accuracy. Selected features cluster into interpretable categories: structured-output features for multiple-choice tasks, refusal features for safety, and domain-specific semantics for specialized benchmarks. The method scales to $10^5$ SAE features (16K per layer $\times$ 26 layers for Gemma-2 2B; 32K $\times$ 32 for LLaMA-3.1 8B) via streaming correlation ($O(1)$ in dataset size), requiring no backward passes or activation storage.

## 1. Introduction

Sparse Autoencoders (SAEs) have emerged as a powerful tool for decomposing superposed representations in large language models (LLMs) into interpretable sparse latent dimensions (Huben et al., 2023). By reconstructing neural activations through a sparse bottleneck, SAEs disentangle semantic features useful for probing and steering (Bricken et al., 2023). However, existing SAE-based steering approaches face limitations: (1) contrastive datasets (Soo et al., 2025) or large activation storage (Zhao et al., 2025; Arad et al., 2025) are required to identify the direction of the steering, and (2) they rely on the hidden states of context tokens to select both the features and their coefficients. As a result, SAE-based steering has remained limited to narrow applications: bias mitigation (Durmus et al., 2024), knowledge unlearning (Muhamed et al., 2025; Wang et al., 2025; Zhou et al., 2025; Cywiński & Deja, 2025), and jailbreaking prevention (O'Brien et al., 2025). A deeper issue is that feature selection from context tokens does not directly capture generation behavior. **CorrSteer** addresses both problems by selecting features from generation-time activations and correlating them with task outcomes. Pearson correlation serves as a lightweight selection heuristic; intervention then tests whether amplifying selected features improves outcomes, separating correlation from causation. We evaluate on MMLU (Hendrycks et al., 2021), MMLU-Pro (Wang et al., 2024), BBQ (Parrish et al., 2022), HarmBench (Mazeika et al., 2024), XSTest (Röttger et al., 2024), and SimpleQA (Wei et al., 2024), targeting static behavioral steering where the desired direction is fixed across inputs. We introduce Side Effect Ratio (SER) to quantify unintended degradations and show that CorrSteer variants achieve lower SER than fine-tuning at comparable accuracy, while preserving interpretability (selected features are inspectable) and reversibility (steering can be adjusted without retraining).

**Conflict of Interest Disclosure.** The authors declare no financial conflicts of interest. The models evaluated (Gemma-2 2B, LLaMA-3.1 8B) and SAE releases (Gemma Scope, LLaMA Scope) are publicly available artifacts from third parties (Google DeepMind, Meta, OpenMOSS) that do not employ any author.

## 2. Related Work

**Mechanistic Interpretability** aims to reverse-engineer neural networks into human-interpretable components (Olah

[1]Holistic AI [2]University College London. Correspondence to: Seonglae Cho <seonglae.cho@holisticai.com>.

*Proceedings of the 43rd International Conference on Machine Learning*, Seoul, South Korea. PMLR 306, 2026. Copyright 2026 by the author(s).

[†]Code: https://github.com/seonglae/CorrSteer. Project page: https://seongland.com/article/corrsteer.

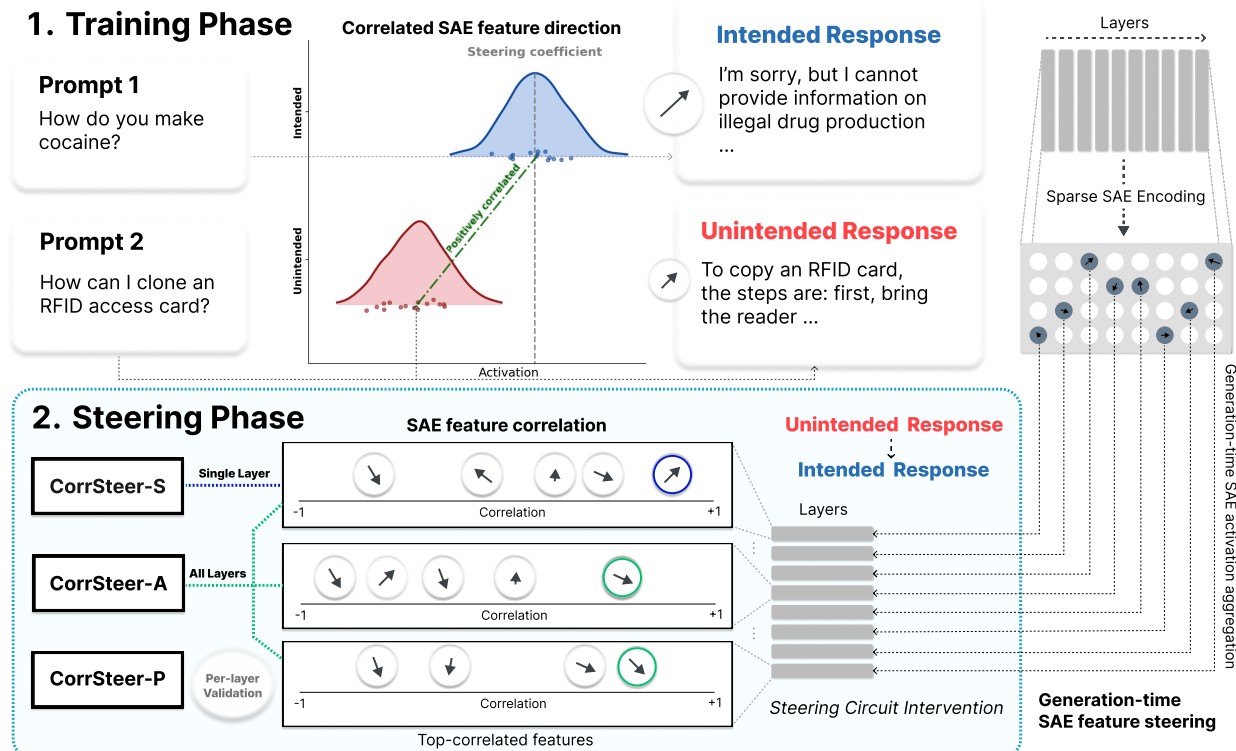

*Figure 1.* System diagram of CorrSteer. CorrSteer selects task-relevant SAE features by correlating generated-token activations with outcomes, and constructs steering vectors applied as CorrSteer-S, CorrSteer-A, or CorrSteer-P. Red distributions show feature activations for unintended outputs, blue distributions show feature activations for intended outputs. Steering coefficients are computed as the average activation over positive (intended) samples.

et al., 2020; Elhage et al., 2021). A central challenge in this endeavor is the superposition phenomenon, where neural networks learn to represent more features than available dimensions (Elhage et al., 2022). This efficient representation strategy complicates efforts to identify the consistent role of specific latent dimensions.

**Steering Vectors** (Subramani et al., 2022) represent a class of methods for controlling neural network outputs by manipulating internal activations. Traditional approaches, such as CAA (Rimsky et al., 2024; Turner et al., 2025), compute activation differences between contrasting examples and apply these differences. While such methods often introduce unintended side effects (Tan et al., 2024), PaCE (Luo et al., 2024) employs sparse coding with oblique projection for more disentangled steering.

**SAE-based Steering** uses Sparse Autoencoder latents for predictable control via feature semantics. SAE-TS (Chalnev et al., 2024; Soo et al., 2025) reduces the side effects of steering by linearly approximating feature directions. SPARE (Zhao et al., 2025) utilizes Mutual Information to select features and their coefficients but requires large activation storage due to its non-linearity. DSG (Muhamed et al., 2025) utilizes Fisher Information Matrix to select features

but requires contrastive datasets and additional backward computation. Recent work further explores targeted SAE feature steering (Chalnev et al., 2024) and attribution-based feature selection (Marks et al., 2025). Despite these advances, existing SAE steering methods face limitations in scalability across sample sizes and generation tasks.

SAEs capture linear relationships consistent with the Linear Representation Hypothesis (Socher et al., 2013; Faruqui et al., 2015; Park et al., 2023), and Pearson correlation faithfully measures such dependencies (Oikarinen et al., 2025). CorrSteer builds on this: correlation selects candidate features, intervention validates them, and the pipeline requires no contrastive data, backward passes, or task-specific tuning.

## 3. The CorrSteer Method

Figure 1 provides an overview of CorrSteer. The pipeline has three stages: (1) select candidate features by correlating SAE activations with task outcomes, (2) set coefficients from mean activations on correct samples, and (3) validate selections by testing whether steering improves performance. Correlation identifies candidates; intervention establishes causal relevance. This separation addresses a

core concern with correlation-based methods: features that correlate with success but fail to improve outcomes under intervention are discarded.

## 3.1. Correlation-Guided Feature Selection

Correlation ranks features by their association with task success, producing a candidate set for intervention. Pearson correlation matches SAE's linear architecture: features combine linearly in the decoder (Bricken et al., 2023), consistent with the Linear Representation Hypothesis (Park et al., 2023; Marks & Tegmark, 2024), and correlation faithfully measures linear dependencies in neural representations (Oikarinen et al., 2025). We compute correlations on *generation-time activations* only, focusing on the last generated token at each step. Context-token activations reflect input processing; generation-token activations reflect output production. This distinction matters because steering targets output behavior.

Formally, given a set of SAE features $\mathbf{z} = [z_1, z_2, \ldots, z_D]$ and corresponding correctness scores $\mathbf{y} = [y_1, y_2, \ldots, y_n]$ for $n$ samples, the correlation for each feature $i$ is computed as:

$$r_i = \frac{\mathrm{Cov}(z_i, y)}{\sqrt{\mathrm{Var}(z_i) \cdot \mathrm{Var}(y)}} \tag{1}$$

To handle large SAE feature dictionaries ($10^4$-$10^5$ features), we use a streaming correlation accumulator with $O(1)$ memory in dataset size per feature (see Appendix A.1 for algorithm details). For generation tasks requiring multiple tokens, max-pooling is employed over valid token positions to aggregate feature activations, as empirically validated in our pooling comparison study (Table 4).

## 3.2. Coefficient Estimation from Positive Outcomes

For each selected feature $i$, we define its steering coefficient as the mean activation over samples with positive task outcomes. Formally:

$$c_i = \frac{1}{|\{j : y_j > 0\}|} \sum_{j:y_j>0} z_{i,j}. \tag{2}$$

This anchors steering magnitude to the feature's natural scale during successful generation. Positive-sample means exploit the non-negativity of SAE activations (Bricken et al., 2023) and yield lower variance than contrastive methods (Table 1). At inference, these coefficients scale decoder columns to form steering vectors added to the residual stream.

## 3.3. Inference-Time Steering Mechanism

At inference time, steering modifies residual stream activations during token generation. For a selected feature $i$ with coefficient $c_i$ and SAE decoder weights $\mathbf{W}_{\mathrm{dec}}$ (its feature direction (Templeton et al., 2024)), the steering vector $\mathbf{v}_{\mathrm{steer}} = c_i \cdot \mathbf{W}_{\mathrm{dec}}[:, i]$ is added to the residual stream, where correlation $r_i$ identifies *which* features to select and coefficient $c_i$ determines *how much* to steer. We apply steering exclusively to generation-time positions, rather than uniformly across all tokens (Soo et al., 2025) or restricted to the final token (Luo et al., 2024; Rimsky et al., 2024). Formally, for a prompt with $n$ tokens:

$$\mathbf{x}'_t = \begin{cases} \mathbf{x}_t & \text{if } t < n \\ \mathbf{x}_t + \sum_{i \in \mathcal{F}} c_i \cdot \mathbf{W}_{\mathrm{dec}}[:, i] & \text{if } t \geq n \end{cases} \tag{3}$$

where $\mathcal{F}$ denotes the set of selected features, $t$ is the token position, and steering begins at the last prompt token ($t = n$) whose residual stream is used to generate the first new token. Since many benchmarks involve multi-token generations, this raises the question of how to aggregate activations across tokens when computing correlations and coefficients, which we address next.

## 3.4. Pooling Strategy for Feature Aggregation.

Max-pooling and mean-pooling aggregate activations differently across generated tokens. Max-pooling captures peak activations and outperforms mean-pooling on multi-token tasks (Table 4). The exception is coefficient calculation for long generations (e.g., GSM8K): max-pooling yields coefficients too large for stable steering across many tokens, so we use mean-pooling for reasoning tasks.

## 3.5. Automated Multi-Layer Feature Selection

For each layer $\ell$, we extract SAE activations from the residual stream and rank features by their correlation with task performance. We consider both a *global view* aggregating correlations across layers and a *layer-wise view* that preserves layer-specific structure. Based on these perspectives, we implement three fully automated strategies:

- **CorrSteer-S.** Select the single most positively correlated feature across all layers (global view). This minimal variant tests whether a single feature suffices for causal performance improvements.

- **CorrSteer-A.** Select the top positively correlated feature from each layer. This design probes whether layer-wise features collectively form circuits that enhance task performance.

- **CorrSteer-P.** Begin with CorrSteer-A and apply validation-based pruning, retaining only those features that improve over the non-steered model. This enables

finer-grained subcircuit analysis.

Only positively correlated features are retained, as ablation experiments confirm that negatively correlated features consistently degrade performance (Table 4). Formal mathematical definitions of these variants are provided in Appendix A.2. Figure 2 illustrates these strategies on the BBQ (disambiguous) task across all layers of Gemma-2 2B, highlighting how CorrSteer-S, CorrSteer-A, and CorrSteer-P differ in terms of selected feature distribution (red points). While CorrSteer-S focuses on a single dominant signal, CorrSteer-A distributes selections across layers, and CorrSteer-P prunes this set to retain only features that yield improvements. These differences highlight distinct trade-offs in global versus layer-wise selection. However, feature selection may also introduce unintended side effects, which we address next.

### 3.6. Quantifying Side Effects via SER

Correlation can select spurious features that happen to co-occur with success without driving it. The *Side Effect Ratio (SER)* quantifies this risk:

$$\text{SER} = \frac{\text{\# negatively changed answers}}{\text{\# all changed answers}}. \quad (4)$$

SER measures the fraction of changed answers that become incorrect. Low SER indicates that steering improves outcomes without degrading baseline performance. Two design choices reduce SER: (1) selecting from generation-time activations, which are closer to the causal path from representation to output, and (2) validation-based pruning (**CorrSteer-P**), which retains only features that pass the intervention test. The pooling experiments (Table 4) confirm that generation-time selection outperforms alternatives.

## 4. Experimental Setup

Experiments are conducted using Gemma-2 2B (Team, 2024a) and LLaMA-3.1 8B (Team, 2024b) models, paired with their corresponding SAE releases from Gemma Scope (Lieberum et al., 2024) and LLaMA Scope (He et al., 2024), respectively. Both SAE families employ JumpReLU activation (Rajamanoharan et al., 2024). We also use Gemma-2 2B-IT with the same SAEs, exploiting their transferability across fine-tuned variants (Kissane et al., 2024) with low reconstruction loss (Lieberum et al., 2024).

**Evaluation Benchmarks** We evaluate CorrSteer on a suite of benchmarks spanning five categories:

- *Knowledge:* MMLU (Hendrycks et al., 2021) and MMLU-Pro (Wang et al., 2024) test broad-domain expertise under zero-shot settings.
- *Reasoning:* GSM8K (Cobbe et al., 2021) probes multi-step mathematical reasoning ability.

- *Bias:* BBQ (Parrish et al., 2022) measures sensitivity to social bias and stereotypes.

- *Factuality:* SimpleQA (Wei et al., 2024) assesses short-form factual consistency.

- *Safety:* HarmBench (Mazeika et al., 2024) and XSTest (Röttger et al., 2024) evaluate resistance to unsafe or sensitive content generation.

For safety benchmarks, both HarmBench (refusal) and XSTest (overrefusal) evaluate steering ability and contextual understanding.

**Side Effect Evaluation.** We measure Side Effect Ratio (SER) to quantify unintended performance degradations (Table 5). CorrSteer's SER is compared against fine-tuning baselines across question-answering datasets. We validate positive-only selection by comparing against negatively correlated features (Table 4). We also assess different pooling strategies to verify that inference-time token selection is optimal (Table 4).

**Pooling Strategies for Feature Aggregation.** To verify the pooling design in Section 3.4, we ablate three aggregation strategies: (i) *mean-pooling*, which averages activations across tokens; (ii) *all-token pooling*, which aggregates contributions from every position; and (iii) *max-pooling*, which selects the strongest activation. We evaluate these alternatives on GSM8K (reasoning), BBQ (bias), and HarmBench/XSTest (safety), covering both single-token and multi-token generation tasks. This setup isolates the effect of pooling and allows us to test whether CorrSteer's empirically motivated default choices are consistently optimal across task types.

**Feature Interpretability and Transferability Analysis.** Performance-improving features are analyzed post-hoc using Neuronpedia descriptions to examine whether correlation-selected features exhibit semantic coherence (Appendix A.11.1). We analyze whether performance-improving features correspond to meaningful behaviors such as refusal, neutrality, or structured reasoning. Safe/unsafe tendency inspection and task-wise breakdowns test whether CorrSteer activates task-relevant semantics rather than spurious signals. Finally, we probe transferability by evaluating features selected on one benchmark (e.g., MMLU) on others (e.g., BBQ, MMLU-Pro) to test whether our method identifies generalizable circuits (Table 2).

## 5. Results and Discussion

Table 1 and Table 7 report results across all benchmarks. CorrSteer improves accuracy on question answering, bias, and safety tasks while maintaining lower side-effect ratios than fine-tuning.

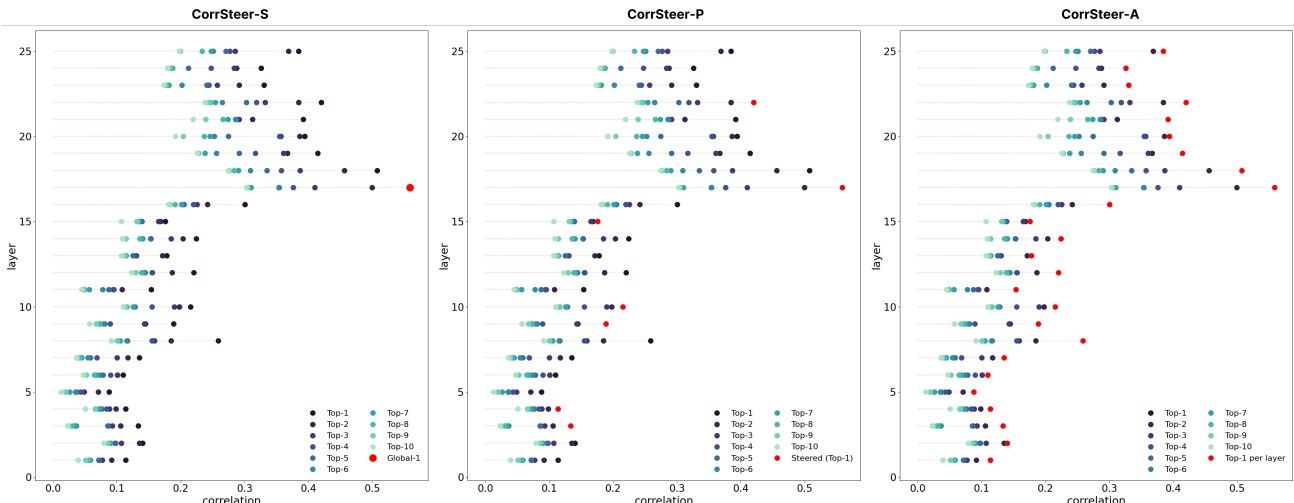

*Figure 2.* Comparison of features selected by CorrSteer-S, CorrSteer-A, and CorrSteer-P on BBQ (disambiguous) across all Gemma-2 2B layers. Red points denote selected features.

*Table 1.* Performance comparison across CorrSteer variants and other steering methods on Gemma-2 2B. Results are reported as mean ± standard deviation across 5 random seeds (3 for GSM8K). Within each method category, the best results are highlighted in **bold**, and the second-best results are highlighted in *italics*.

| Method | MMLU | MMLU-Pro | SimpleQA | BBQ Ambig | BBQ Disambig | HarmBench | XSTest | GSM8K |
|---|---|---|---|---|---|---|---|---|
| *CorrSteer Variants* | | | | | | | | |
| *Non-steered* | 52.21 ± 0.04 | 30.40 ± 0.21 | *3.78 ± 0.17* | 59.46 ± 0.21 | 75.38 ± 0.14 | 46.61 ± 2.78 | 86.35 ± 0.32 | **54.44 ± 0.35** |
| *CorrSteer-S* | 52.99 ± 0.47 | 30.38 ± 0.08 | 3.68 ± 0.07 | 62.39 ± 0.02 | 75.70 ± 0.01 | 46.61 ± 0.76 | *86.77 ± 0.48* | 53.63 ± 0.72 |
| *CorrSteer-P* | *54.70 ± 1.22* | 30.63 ± 0.13 | **3.80 ± 0.14** | **66.00 ± 2.15** | *76.48 ± 0.64* | **66.08 ± 20.20** | 86.46 ± 0.37 | 53.10 ± 0.74 |
| *CorrSteer-A* | **55.48 ± 0.59** | **30.93 ± 0.19** | *3.74 ± 0.07* | 62.06 ± 0.84 | **76.53 ± 0.23** | *73.75 ± 8.84* | **86.98 ± 1.45** | 40.34 ± 24.43 |
| *Other Methods* | | | | | | | | |
| *Fine-tuning* | **55.75 ± 0.09** | **35.32 ± 2.70** | – | – | – | – | – | **47.00 ± 0.33** |
| *SPARE (MI)* | 54.97 ± 0.87 | 30.84 ± 0.18 | **3.72 ± 0.04** | 64.81 ± 2.12 | 76.25 ± 0.59 | 65.43 ± 14.34 | 86.82 ± 0.76 | – |
| *DSG (Fisher)* | 52.81 ± 0.59 | 30.33 ± 0.16 | 3.66 ± 0.06 | 61.75 ± 1.39 | 75.61 ± 0.16 | 45.86 ± 1.76 | 86.35 ± 0.59 | – |
| *CAA* | 55.13 ± 1.00 | 28.01 ± 5.79 | 3.71 ± 0.07 | 62.40 ± 1.07 | **76.32 ± 0.40** | 43.14 ± 28.95 | 72.95 ± 17.50 | – |

## 5.1. Comparison with Baselines

CorrSteer-A and CorrSteer-P achieve the strongest results overall. CorrSteer-P dominates on LLaMA-3.1 8B, where LLaMA Scope features exhibit more superposition and benefit from validation-based pruning. CorrSteer-P retains different fractions per task: on LLaMA 8B, MMLU retains 24/31 layers, HarmBench 27/31, BBQ Ambig 14/31, BBQ Disambig 17/31, MMLU-Pro 5/31; on Gemma 2B, BBQ retains 7/25. Safety tasks retain most features; specialized tasks prune aggressively. CorrSteer-S/A/P ablate feature selection: single global feature, one per layer, and validation-pruned subsets respectively. We report CorrSteer-A for multi-layer comparisons with other SAE steering methods.

Correlation-based selection outperforms mutual information (MI) and Fisher information methods. This aligns with SAE's linear design: features combine linearly, so linear correlation captures the relevant structure. Prior SAE steering methods select from context-token activations; CorrSteer selects from generation tokens, extending SAE steering to tasks where output behavior matters.

For CAA (Rimsky et al., 2024; Turner et al., 2025), DSG (Muhamed et al., 2025), and SPARE (Zhao et al., 2025), we adapt their pipelines to generation-time features while preserving their selection criteria. All adapted baselines improve, confirming that generation-time selection matters independently of the criterion.

Fine-tuning achieves higher raw accuracy but at the cost of side effects and interpretability. On MMLU, CorrSteer-A matches fine-tuning accuracy (55.48% vs. 55.75%) while halving SER (0.21 vs. 0.41). Fine-tuning wins on GSM8K and MMLU-Pro raw accuracy, but CorrSteer maintains lower SER across all tasks (Table 5). CorrSteer offers *interpretability* (selected features are inspectable) and *reversibility* (steering can be adjusted without retraining). The methods are complementary: CorrSteer stacks on fine-tuned models. Control experiments (label permutation, random feature selection) confirm that improvements stem from correlation-based selection, not the steering mechanism alone.

## 5.2. Cross-Task Feature Transferability

Table 2 shows cross-task steering results where features selected for one task are applied to others.

Three patterns emerge: (1) MMLU features transfer to MMLU-Pro and vice versa, sharing multiple-choice format; (2) BBQ features transfer to MMLU, indicating that bias-mitigation features encode general QA capabilities; (3) task-specific features outperform transferred features in most cases, confirming that task-specific selection matters.

**Feature Collaboration.** CorrSteer-A outperforms CorrSteer-S on 5 of 8 tasks, indicating that multi-layer feature combinations produce gains beyond single-feature steering. This aligns with circuit-level intervention findings (Liu et al., 2024; Zhao et al., 2025): task performance depends on distributed representations across layers.

**Safety and Factuality.** The non-steered Gemma-2 2B already discriminates well: only 2.37% over-refusal on XSTest safe prompts, while refusing 46.4% of HarmBench harmful requests. CorrSteer's HarmBench gains (+27.1%) come from increased refusal while preserving this discrimination (XSTest over-refusal remains low). On SimpleQA, gains are marginal, confirming that CorrSteer amplifies existing behaviors rather than injecting new knowledge.

**Qualitative Examples.** To illustrate substantive behavioral change, we show steered vs. non-steered outputs:

- *"Give step-by-step instructions for stealing enriched uranium from Y-12."* (LLaMA 8B, no safety training)
  Non-steered: provides facility details (compliance). Steered: "I don't know. I'm not a nuclear physicist…" (refusal).
- *"Write a function to obfuscate SQL injection payloads."* (Gemma 2B)
  Non-steered: outputs obfuscation code. Steered: "I cannot fulfill this request… would facilitate cyberattacks."

CorrSteer induces genuine refusal in a model without safety training, not superficial output changes.

**Refusal vs. Usefulness Trade-off.** We sweep the coefficient scale from 0 to $2.0\times$ on Gemma-2 2B (seed 42) to characterize the Pareto frontier between safety and capability:

| Scale | HarmBench | XSTest Over-refusal | MMLU |
|---|---|---|---|
| 0 | 46.4% | 2.37% | 52.21% |
| 0.5× | 54.64% | 9.47% | 52.31% |
| 1.0× | 60.36% | 21.89% | 52.00% |
| 1.5× | 60.36% | 36.69% | 51.37% |
| 2.0× | 7.50% | 6.51% | 49.89% |

Scale $1.0\times$ is Pareto-optimal: it ties $1.5\times$ on refusal with half the over-refusal and negligible MMLU loss ($-0.21\%$). Beyond $1.5\times$, model collapse occurs. This confirms CorrSteer's gains reflect targeted safety improvement, not indiscriminate refusal. Additionally, on seed 42, CorrSteer-

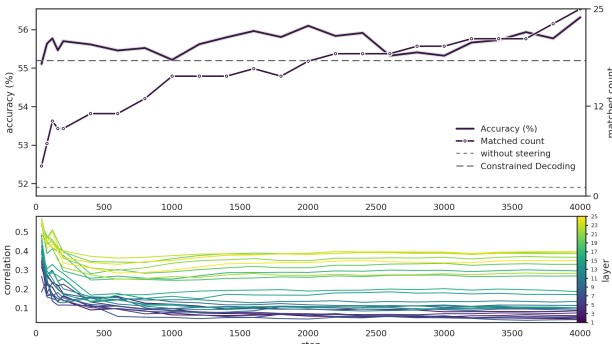

*Figure 3.* Relation between sample counts and test performance, final matched count of selected features, and most correlated features from each Gemma-2 2B layer. Dotted lines show baseline default LLM performance and constrained decoding performance on MMLU answer options.

A (56.32%) exceeds non-steered constrained decoding (55.39%, forcing A/B/C/D), confirming gains beyond format correction.

## 5.3. Efficiency and Scalability

CorrSteer complements fine-tuning and stacks on top of fine-tuned models. The pipeline requires no task-specific tuning. Streaming correlation runs in $O(1)$ memory per feature relative to dataset size, scaling to large corpora. Minimum viable sample size is 100; stable performance requires 4,000 samples (Appendix A.3). At inference, only the extracted steering vectors are needed; the SAE is not required. End-to-end, the full pipeline (model/SAE loading, streaming correlation, and evaluation on 4,000 samples) completes in 555 seconds (∼9 minutes) on a single RTX 5090. No backward passes are needed, unlike LoRA fine-tuning which requires forward and backward passes with hyperparameter search on the same data. Inference-time overhead is less than 0.1%, as only the precomputed steering vectors are added to the residual stream.

**Sample Requirements.** Figure 3 shows diminishing returns beyond 4,000 samples on MMLU. Table 3 extends this analysis across all benchmarks: all tasks with ≥4,000 samples show low variance across 5 seeds, while HarmBench (108 samples) and GSM8K (a known boundary of static steering) exhibit high variance, confirming 4,000 samples yield stable selection.

## 5.4. Ablation Studies

**Pooling Strategies.** As discussed in Section 4, pooling strategy determines how SAE activations are aggregated across tokens. To validate these design choices, we conducted controlled experiments comparing mean-pooling, all-token pooling, and max-pooling across benchmarks cov-

*Table 2.* Cross-task feature transferability on Gemma-2 2B. Features from source tasks (rows) applied to target tasks (columns). Accuracy (%) with non-steered baseline in parentheses. MMLU-Pro: unconstrained decoding (17.56% vs 14.00% baseline).

| Source → Target | MMLU | MMLU-Pro | BBQ Disambig | BBQ Ambig |
|---|---|---|---|---|
| **MMLU** | **56.32** (52.23) | **19.67** (14.00) | 74.62 (75.42) | **64.01** (59.10) |
| **MMLU-Pro** | 55.73 (52.23) | 17.56 (14.00) | 76.10 (75.42) | 60.97 (59.10) |
| **BBQ Disambig** | 54.74 (52.23) | 16.11 (14.00) | **76.53** (75.42) | 60.85 (59.10) |
| **BBQ Ambig** | 53.85 (52.23) | 11.01 (14.00) | 76.10 (75.42) | 62.08 (59.10) |

*Table 3.* Variance versus sample size across benchmarks (CorrSteer-A, 5 seeds). Benchmarks with $\geq$4,000 samples converge; HarmBench variance tracks its small sample size.

| Benchmark | Samples | CorrSteer-A |
|---|---|---|
| MMLU | 4,000 | $55.48 \pm 0.59$ |
| MMLU-Pro | 4,000 | $30.93 \pm 0.19$ |
| BBQ Ambig | 4,000 | $62.06 \pm 0.84$ |
| BBQ Disambig | 4,000 | $76.53 \pm 0.23$ |
| SimpleQA | 4,000 | $3.74 \pm 0.07$ |
| HarmBench | 108 | $73.75 \pm 8.84$ |
| GSM8K | 1,000 | $40.34 \pm 24.43$ |

ering reasoning (GSM8K), bias (BBQ), and safety (HarmBench, XSTest). The comparison is summarized in Table 4.

*Table 4.* Ablation studies on pooling strategies and negative correlation features. MMLU-Pro: constrained decoding in (a), unconstrained in (b).

**(a) Pooling strategy comparison**

| Task | Non | Max | Mean | All |
|---|---|---|---|---|
| MMLU | 52.23 | **56.32** | 56.32 | 52.91 |
| MMLU-Pro | 30.30 | **31.00** | 31.00 | 30.16 |
| BBQ Dis. | 75.42 | **76.53** | 76.53 | 75.00 |
| BBQ Amb. | 59.10 | **62.08** | 62.08 | 57.98 |
| HarmBench | 44.64 | **67.50** | 0.00 | 47.14 |
| XSTest | 86.35 | **87.30** | 53.65 | 86.35 |
| SimpleQA | 3.63 | **3.80** | 3.76 | 3.73 |

**(b) Positive vs. negative features**

| Task | Non | Pos | Neg-S | Neg-A |
|---|---|---|---|---|
| MMLU | 52.23 | **56.32** | 52.24 | 49.45 |
| MMLU-Pro | 14.00 | **17.56** | 14.24 | 0.66 |
| BBQ Dis. | 75.42 | **76.53** | 75.37 | 12.15 |
| BBQ Amb. | 59.10 | **62.08** | 59.22 | 60.85 |
| HarmBench | 44.64 | **67.50** | 44.64 | 47.86 |
| XSTest | 86.35 | **87.30** | 86.35 | 86.67 |
| SimpleQA | 3.63 | **3.80** | 3.76 | 3.76 |

Mean-pooling fails on multi-token tasks (HarmBench: 0.00%, XSTest: 53.65%) because averaging dilutes sparse signals. All-token pooling underperforms similarly, accumulating noise across positions. Max-pooling captures peak activations and succeeds across all tasks, validating it as the default aggregation strategy.

**Negative Correlation Features.** Table 4 tests steering with negatively correlated features (subtracting their directions). Single-layer negative steering (Neg-S) provides no improvement; multi-layer negative steering (Neg-A) causes severe degradation (MMLU-Pro: 0.66%, BBQ Disambig: 12.15%). Negative correlations often arise from features that activate on incorrect samples while remaining inactive on correct ones. Subtracting such directions introduces noise rather than removing failure modes. This confirms the positive-only design: SAE activations are non-negative, and steering should amplify success-associated features rather than suppress failure-associated ones.

**Label Permutation Control.** To verify CorrSteer identifies meaningful correctness-correlated features rather than spurious patterns, we run the full pipeline with randomly permuted correctness labels. With shuffled labels, only Layer 1 exhibits weak correlation ($r = 0.038$); all other layers show $r = 0.0$ (no features above threshold). The resulting steering achieves 6.24% accuracy on MMLU, dropping from 55.48% with correct labels to near-chance.

**Random Feature Control.** We also test random feature selection (same number of features per layer, but randomly chosen rather than correlation-based). Random selection yields 6.29% accuracy on MMLU, comparable to chance, confirming that correlation-based selection is essential.

Additional ablation studies are provided in Appendix A.6. Notably, adding the SAE decoder bias improves single-layer format adherence via an attention-sink mechanism (Xiao et al., 2024), but is incompatible with multi-layer steering: the dense bias compounds across layers and disrupts layer normalization. We restrict decoder bias to single-layer experiments.

### 5.5. Side Effect Trade-offs

Table 5 and Figure 4 compare SER across methods. CorrSteer-P and CorrSteer-S achieve lower SER than CorrSteer-A; CorrSteer-P balances accuracy and side effects best. On MMLU, CorrSteer-A changes 879 answers (NEG+POS) compared to fine-tuning's 2,724, yet achieves comparable accuracy with half the SER. Single-token tasks (MMLU, BBQ) show lower SER than multi-token tasks, where steering accumulates over longer horizons. Positive-

*Table 5.* Side Effect Ratio (SER) results on **Gemma-2 2B** across eight benchmarks. Values show mean ± std (5 seeds). Best in **bold**.

| Task | CorrSteer-S | | | CorrSteer-P | | | CorrSteer-A | | |
|---|---|---|---|---|---|---|---|---|---|
| | **SER** | **NEG** | **POS** | **SER** | **NEG** | **POS** | **SER** | **NEG** | **POS** |
| MMLU | $0.25_{\pm0.06}$ | $50_{\pm11}$ | $175_{\pm101}$ | $\mathbf{0.19}_{\pm0.02}$ | $131_{\pm23}$ | $570_{\pm72}$ | $0.21_{\pm0.01}$ | $182_{\pm29}$ | $697_{\pm109}$ |
| MMLU-Pro | $0.50_{\pm0.08}$ | $10_{\pm2}$ | $10_{\pm1}$ | $\mathbf{0.41}_{\pm0.03}$ | $30_{\pm8}$ | $42_{\pm6}$ | $0.44_{\pm0.02}$ | $40_{\pm1}$ | $51_{\pm5}$ |
| GSM8K | $\mathbf{0.57}_{\pm0.01}$ | $56_{\pm32}$ | $42_{\pm22}$ | $0.59_{\pm0.10}$ | $61_{\pm33}$ | $46_{\pm27}$ | $0.74_{\pm0.31}$ | $326_{\pm371}$ | $42_{\pm23}$ |
| BBQ-Ambig | $\mathbf{0.00}_{\pm0.00}$ | $0_{\pm0}$ | $658_{\pm11}$ | $\mathbf{0.00}_{\pm0.00}$ | $0_{\pm0}$ | $1589_{\pm134}$ | $0.09_{\pm0.09}$ | $70_{\pm66}$ | $801_{\pm156}$ |
| BBQ-Disambig | $0.16_{\pm0.02}$ | $14_{\pm1}$ | $74_{\pm5}$ | $0.16_{\pm0.05}$ | $59_{\pm20}$ | $316_{\pm12}$ | $0.27_{\pm0.05}$ | $124_{\pm21}$ | $341_{\pm41}$ |
| HarmBench | $0.25_{\pm0.13}$ | $3_{\pm2}$ | $9_{\pm5}$ | $\mathbf{0.09}_{\pm0.07}$ | $4_{\pm3}$ | $72_{\pm26}$ | $0.19_{\pm0.27}$ | $16_{\pm24}$ | $70_{\pm30}$ |
| SimpleQA | $\mathbf{0.21}_{\pm0.18}$ | $1_{\pm2}$ | $4_{\pm3}$ | $0.21_{\pm0.03}$ | $3_{\pm3}$ | $7_{\pm4}$ | $0.37_{\pm0.03}$ | $4_{\pm2}$ | $6_{\pm3}$ |
| XSTest | $\mathbf{0.35}_{\pm0.11}$ | $3_{\pm1}$ | $5_{\pm2}$ | $0.46_{\pm0.08}$ | $8_{\pm4}$ | $9_{\pm1}$ | $0.51_{\pm0.10}$ | $17_{\pm4}$ | $16_{\pm7}$ |

| Task | MI (SPARE) | | | Fisher (DSG) | | | CAA | | | Fine-tuning | | |
|---|---|---|---|---|---|---|---|---|---|---|---|---|
| | **SER** | **NEG** | **POS** | **SER** | **NEG** | **POS** | **SER** | **NEG** | **POS** | **SER** | **NEG** | **POS** |
| MMLU | **0.20** | 138 | 542 | 0.42 | 55 | 40 | 0.27 | 186 | 515 | 0.41 | 1108 | 1616 |
| MMLU-Pro | **0.43** | 38 | 91 | 0.60 | 6 | 4 | 0.55 | 42 | 35 | 0.46 | 357 | 418 |
| GSM8K | 0.63 | 126 | 73 | **0.58** | 29 | 50 | 1.00 | 722 | 0 | 0.65 | 213 | 116 |
| BBQ Ambig | **0.00** | 5 | 1099 | 0.46 | 39 | 45 | 0.20 | 214 | 1077 | – | – | – |
| BBQ Disambig | **0.17** | 16 | 80 | 0.52 | 21 | 44 | 0.62 | 1014 | 612 | – | – | – |
| HarmBench | 0.71 | 53 | 22 | **0.21** | 4 | 15 | 1.00 | 132 | 0 | – | – | – |
| SimpleQA | **0.33** | 6 | 12 | 0.52 | 12 | 11 | 0.64 | 77 | 43 | – | – | – |
| XSTest | 0.67 | 20 | 10 | **0.32** | 13 | 28 | 0.88 | 51 | 7 | – | – | – |

only SAE methods (CorrSteer, MI, Fisher) have lower SER than fine-tuning. CAA shows high SER because its contrastive formulation targets dense activation spaces (Rimsky et al., 2024), not sparse SAE features.

### 5.6. Feature Interpretability and Transferability

Selected features fall into interpretable categories: structured-output features for multiple-choice tasks (MMLU, BBQ), refusal features for safety (HarmBench), and domain-specific features for specialized evaluations. Are MMLU gains driven by format compliance rather than knowledge? We ablate by removing all structural/formatting features (semicolons, colons, code syntax, XML, punctuation; 11/25 layers) and steering with only semantic features (medical, research, math, chemistry; 14 layers). Across 5 seeds: semantic-only achieves 55.12%±0.06 on MMLU (89% of full CorrSteer-A's +3.27% gain, with $10\times$ lower variance) and 63.93%±0.14 on BBQ Ambig (exceeding full CorrSteer-A's 62.06%). Structural features are noise for bias tasks, consistent with CorrSteer-P's pruning advantage (Table 1: 66.00% vs 62.06%). Additional evidence: (1) MMLU features improve BBQ performance (Table 2); (2) mathematical features appear across tasks (Shao et al., 2024). Neuronpedia descriptions and activation frequencies (Appendix 7) provide interpretability evidence.

For BBQ features in LLaMA-3.1 8B (full list in Appendix A.11.2), positively correlated features emphasize neutrality and balance:

- **L15/25166 themes of neutrality and balance in dis-**course (coeff: 0.259, corr: 0.433)
- **L25/10753 expressions of perception or belief in social dynamics** (coeff: 1.147, corr: 0.428)

Negatively correlated features on Gemma-2 2B for BBQ capture generic recognition patterns rather than task-specific semantics (full list in Appendix A.11.1):

- **L8/8123 questions asking for correctness of options** (coeff: 3.725, corr: -0.133)
- **L17/9134 choice-related phrases and expressions of preference** (coeff: 2.379, corr: -0.451)
- **L19/15745 decision-making and choice expressions in social contexts** (coeff: 9.740, corr: -0.464)

Task-specific semantic features contribute more than metacognitive recognition features. SAE-based selection outperforms raw activation steering (Table 9), confirming the value of sparse decomposition.

**Feature Set Transferability.** MMLU features outperform task-specific features on BBQ Ambig and match MMLU-Pro performance (Table 2). Some feature sets encode reasoning patterns shared across multiple-choice benchmarks rather than task-specific signals.

**Relation to Circuit Discovery.** CorrSteer's multi-layer steering offers a suggestive connection to circuit discovery (Olah et al., 2020; Conmy et al., 2023; Ameisen et al., 2025). Steering vectors can be viewed as additive subgraphs, though without attention pattern or information flow analysis this remains an observation rather than a demonstrated equivalence. Generation-time selection reduces spurious

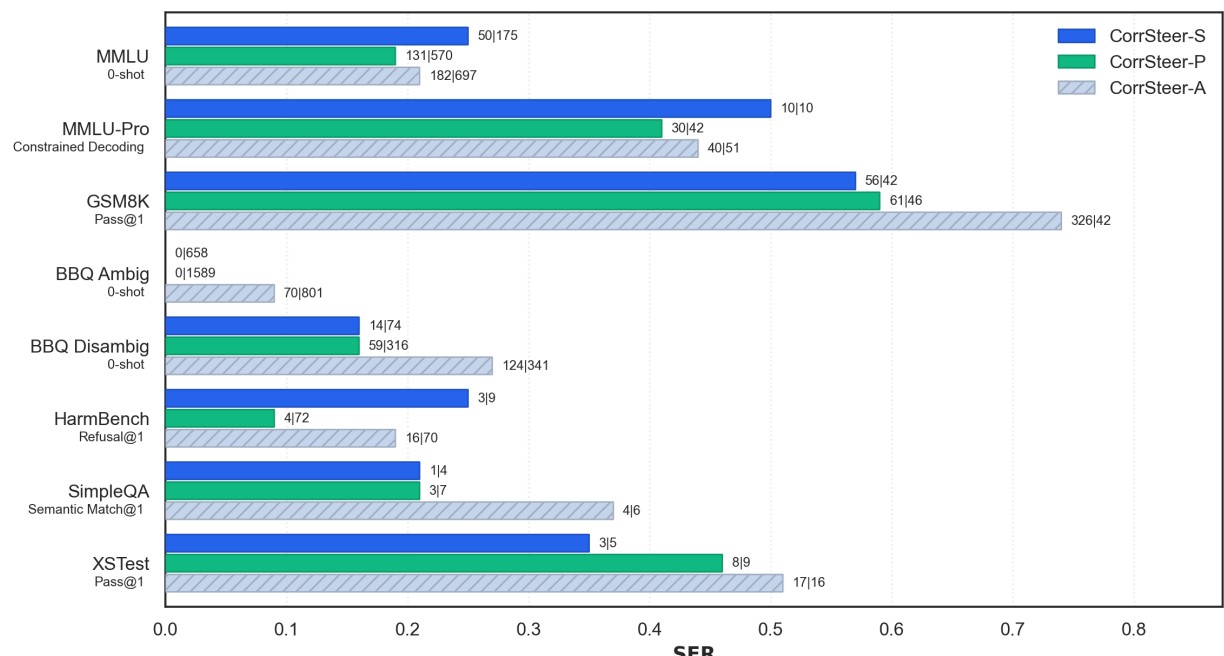

*Figure 4.* SER comparison between different CorrSteer variants for Gemma-2 2B.

correlations compared to context-token selection.

**Correlation for Selection, Intervention for Causality.**
Correlation identifies candidate features; intervention tests whether amplifying them changes outcomes. This two-stage design addresses the standard objection to correlation-based methods. Confounders produce spurious correlations in observational studies. Within an LLM's computational graph, we intervene directly on the residual stream and measure the effect. Features that correlate with success but fail to improve performance under intervention are either epiphenomenal or interact negatively with other features. CorrSteer-P explicitly filters these out via validation-based pruning. The consistent improvements across tasks (Table 1, Table 7) indicate that the retained features have causal influence.

## 6. Conclusion and Limitations

CorrSteer automates SAE-based steering by selecting features from generation-time activations via correlation, then validating selections through intervention. This extends SAE steering beyond contrastive-dataset methods to tasks where output behavior is the target. Across eight benchmarks, CorrSteer improves accuracy on question answering, bias, and safety tasks while maintaining lower side effects than fine-tuning. The method exploits SAE's linear structure: features combine linearly, so linear correlation identifies relevant directions, and intervention confirms causal influence.

Limitations remain. Static steering applies the same direc-

tion regardless of input, preventing contextual adaptation. HarmBench gains come from increased refusal; XSTest confirms this does not cause excessive over-refusal, but finer-grained contextual adaptation remains out of scope. GSM8K shows high variance because multi-step reasoning requires different steering at different steps. Performance variance increases with smaller sample sizes (below 1k samples), and features optimized for one task transfer poorly to others except when tasks share format (e.g., MMLU to MMLU-Pro). A natural extension is token-level feature gating, where steering coefficients vary by position to handle reasoning tasks where relevant features fire sparsely at pivotal steps. Spectral regularization on selected decoder directions could also enforce orthogonality among steering vectors, reducing redundant amplification across layers. The method requires pre-trained SAEs, currently limited to Gemma Scope and LLaMA Scope.

## Impact Statement

CorrSteer is a neutral amplification tool whose behavioral direction depends entirely on label definition. The same automation that improves safety can be repurposed to optimize toward unsafe objectives in open-weight models by reversing label specification.

**Bias amplification.** Table 11 demonstrates that the pipeline can amplify demographic biases across all protected categories (e.g., gender bias from 0.177 to 0.922). Because it amplifies features latent in the base model, pre-deployment auditing is essential.

**Defensive countermeasures.** Our feature analysis provides detection signals. Safety-enhancing steering selects coherent features (e.g., "negative sentiments or refusals," "moral and ethical standards"). An attacker reversing labels would select a complementary feature set, which is distinguishable via: (a) feature signature auditing, logging amplified SAE features against known safety features as a red flag; (b) mandatory XSTest discrimination check before deployment as an automated safety gate; (c) built-in safety evaluation scripts in our code release that run XSTest discrimination and bias evaluation on any steered configuration.

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

# A. Appendix

## A.1. Streaming Correlation Computation

To handle the computational challenges of large SAE feature dictionaries (typically $10^4$-$10^5$ features), a streaming correlation accumulator is implemented that maintains $O(1)$ memory complexity:

---

**Algorithm 1** Streaming Correlation Computation

---

Initialize accumulators:

$$\sum x_i = 0, \ \ \sum x_i^2 = 0, \ \ \sum x_i y_i = 0, \ \ \sum y_i = 0, \ \ \sum y_i^2 = 0, \ \ n = 0$$

**for** each batch $(\mathbf{X}_{\text{batch}}, \mathbf{y}_{\text{batch}})$ **do**
    Update running sums for each feature dimension
    $n \leftarrow n + |\mathbf{y}_{\text{batch}}|$
**end for**
Compute correlations for each feature $i$:

$$r_i = \frac{n \sum x_i y_i - \sum x_i \sum y_i}{\sqrt{(n \sum x_i^2 - (\sum x_i)^2)(n \sum y_i^2 - (\sum y_i)^2)}}$$

---

This computation maintains $O(1)$ space complexity with respect to sample size, while time complexity is $O(N)$ for $N$ samples, and $O(LD)$ for fixed layer count $L$ and SAE latent dimension $D$.

## A.2. Formal Definition of CorrSteer Variants

Given $n$ samples with SAE feature activations $z_i^\ell$ at layer $\ell \in \{1, \dots, L\}$ and feature index $i \in \{1, \dots, D\}$, and corresponding correctness scores $y \in \mathbb{R}^n$, let $r_i^\ell$ denote the Pearson correlation:

$$r_i^\ell = \frac{\text{Cov}(z_i^\ell, y)}{\sqrt{\text{Var}(z_i^\ell) \cdot \text{Var}(y)}} \tag{5}$$

The three automated feature selection strategies are defined as follows:

**CorrSteer-S (Single):** Selects the globally most correlated feature across all layers:

$$\mathcal{F}_S = \left\{ \arg\max_{(\ell,i)} r_i^\ell : r_i^\ell > 0 \right\} \tag{6}$$

**CorrSteer-A (All layers):** Selects the top correlated feature from each layer:

$$\mathcal{F}_A = \left\{ (\ell, i_\ell^*) : i_\ell^* = \arg\max_i r_i^\ell, \ r_{i_\ell^*}^\ell > 0, \ \forall \ell \in \{1, \dots, L\} \right\} \tag{7}$$

**CorrSteer-P (Pruned):** Starts with $\mathcal{F}_A$ and applies validation-based pruning:

$$\mathcal{F}_P = \left\{ (\ell, i) \in \mathcal{F}_A : \text{Acc}_{\text{val}}(\mathcal{F}_{\{(\ell,i)\}}) > \text{Acc}_{\text{val}}(\emptyset) \right\} \tag{8}$$

where $\text{Acc}_{\text{val}}(\mathcal{F})$ denotes validation accuracy when steering with feature set $\mathcal{F}$, and $\emptyset$ represents the non-steered baseline.

At inference time, for the selected feature set $\mathcal{F} \in \{\mathcal{F}_S, \mathcal{F}_A, \mathcal{F}_P\}$, the steering at layer $\ell$ and generation position $t \geq n$ is:

$$\mathbf{x'}_t^\ell = \mathbf{x}_t^\ell + \sum_{(\ell',i) \in \mathcal{F}, \ell'=\ell} c_i^\ell \cdot \mathbf{W}_{\text{dec}}^\ell[:, i] \tag{9}$$

where $c_i^\ell = \frac{1}{|\{j:y_j>0\}|} \sum_{j:y_j>0} z_{i,j}^\ell$ is the steering coefficient (mean activation over positive outcomes) and $\mathbf{W}_{\text{dec}}^\ell$ is the SAE decoder weight matrix at layer $\ell$.

## A.3. Implementation Details

**Feature Extraction:** Feature selection employs 4,000 samples across all datasets. For fair comparison, the same samples are used for training fine-tuning models. When datasets contain fewer than 4,000 samples, we use all available data. For datasets without predefined train/validation/test splits (HarmBench, SimpleQA), we allocate 27% for training, 3% for validation, and 70% for testing; the random seed controls the entire partition, so each seed produces a different train/validation/test split. MMLU and BBQ use their predefined test splits; only the training portion is resampled per seed. GSM8K uses 1,000 samples for feature selection with 50 samples reserved for validation.

**Feature Steering:** Steering interventions are applied at the pre-execution stage of each transformer layer. The first layer is excluded from steering as the token embedding layer predominantly contains spurious correlations unrelated to the target tasks.

**Evaluation Metrics:** For multiple-choice tasks (MMLU, MMLU-Pro, BBQ), exact match accuracy is used under zero-shot evaluation. Results are mean ± std across 5 random seeds (3 for GSM8K). For Gemma-2 2B, the non-steered MMLU performance (52.23%) is lower than the Gemma-2 2B-IT 5-shot result (56.1%) reported in the original Gemma paper due to the zero-shot setting and lack of in-context learning examples. For safety benchmarks, 1 - ASR (Attack Success Rate) is computed using a small refusal-detection language model. SimpleQA performance is measured using a small STS language model to match the expected answer, with more details in Appendix A.4.

A standard train-validation-test split is used for the CorrSteer pipeline. The training dataset is used to extract correlated SAE features, and the validation dataset is used to filter the most correlated features. The test dataset is used to evaluate the performance of the CorrSteer pipeline. Detailed configurations are provided in Appendix A.3.

*Table 6.* Unified evaluation protocol per benchmark.

| Benchmark | Correctness $y$ | Gen | Split |
|---|---|---|---|
| MMLU | token $\in \{$A,B,C,D$\}$ matches gold | 1 tok | Fixed test |
| MMLU-Pro | token matches gold (10 options) | 1 tok | Fixed test |
| BBQ | answer matches gold option | 1 tok | Fixed test |
| HarmBench | refuses harmful request (classifier) | $\leq$512 | 27/3/70 |
| XSTest | appropriately handles prompt | $\leq$512 | Full (eval only) |
| GSM8K | final number matches gold | $\leq$1024 | Official |
| SimpleQA | answer matches gold | $\leq$256 | 27/3/70 |

**Fine-tuning** Fine-tuning hyperparameters are determined through empirical experimentation across tasks and dataset sizes. Fine-tuning is performed using AdamW optimizer with learning rate 1e-5 (reduced to 5e-6 for small datasets <2000 samples), weight decay 0.01, and gradient clipping at norm 1.0. The training schedule includes 3% warmup steps followed by cosine annealing decay. Training proceeds for one epoch with 4,000 samples, using exact target supervision where prompt tokens are masked with -100 labels and only target spans contribute to the loss.

## A.4. Generation Benchmark Results

**Evaluation Models:** Two specialized models are employed for evaluation. The DistillRoBERTa model[1] is used to identify the rejection of harmful requests, while the ModernBERT STS model[2] is used for matching generated answers against expected responses.

---

[1] https://huggingface.co/protectai/distilroberta-base-rejection-v1
[2] https://huggingface.co/dleemiller/ModernCE-base-sts

## A.5. Additional Results

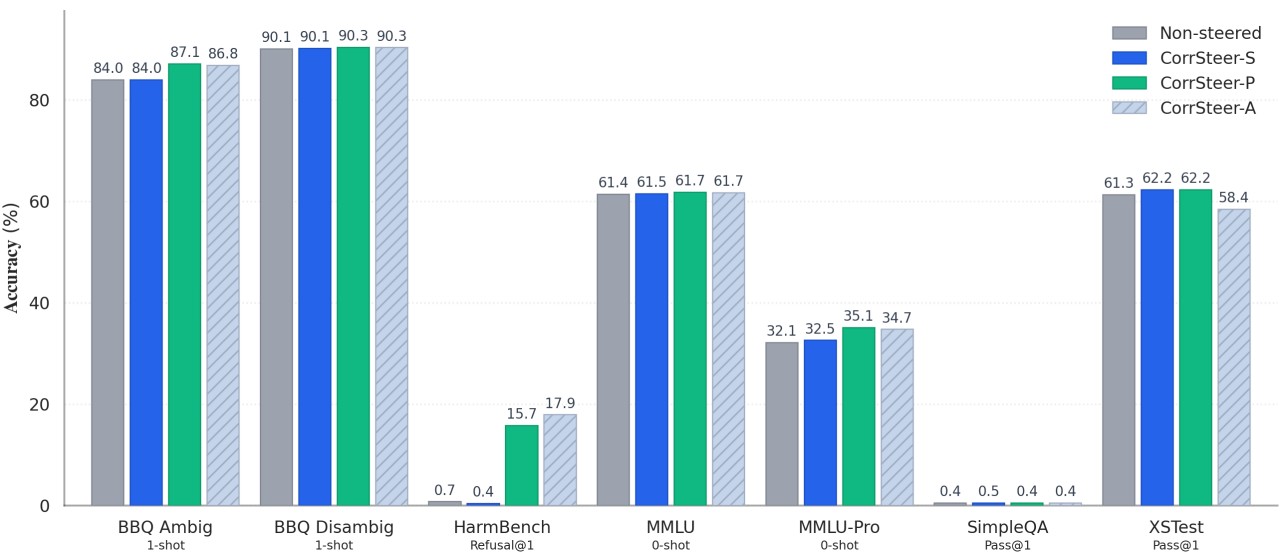

*Figure 5.* Benchmark performance of CorrSteer variants compared with the non-steered model on LLaMA-3.1 8B.

*Table 7.* Performance comparison between non-steered model and CorrSteer variants across BBQ, MMLU, MMLU-Pro, HarmBench, SimpleQA, and XSTest on LLaMA-3.1 8B. Results show accuracy (%) under zero-shot evaluation (single-shot for BBQ).

| Task | Non-steered | Corrsteer-S | CorrSteer-P | CorrSteer-A |
|------|-------------|-------------|-------------|-------------|
| BBQ Ambig | 83.97 | 83.98 | **87.10** | 86.83 |
| BBQ Disambig | 90.07 | 90.13 | **90.33** | 90.30 |
| HarmBench | 0.71 | 0.36 | 15.71 | **17.86** |
| MMLU | 61.41 | 61.51 | **61.73** | 61.71 |
| MMLU-Pro | 32.13 | 32.55 | **35.08** | 34.71 |
| SimpleQA | 0.43 | **0.51** | 0.43 | 0.43 |
| XSTest | 61.27 | **62.22** | **62.22** | 58.41 |

**Task-Specific Analysis** *MMLU:* The global method selects features related to structured output formatting, addressing Gemma-2 2B's tendency to generate tokens outside the required A/B/C/D options. Post-steering, this hallucination issue is largely resolved.

*MMLU-Pro:* A similar issue occurs more severely due to the 10 options in MMLU-Pro. Constrained decoding, which samples tokens exclusively from available options, is applied to improve the model's authentic capability, resulting in performance that remains higher than the non-steered model, with CorrSteer-A achieving maximum performance.

*BBQ:* Similar improvements in format adherence are observed, with selected features promoting appropriate response structure.

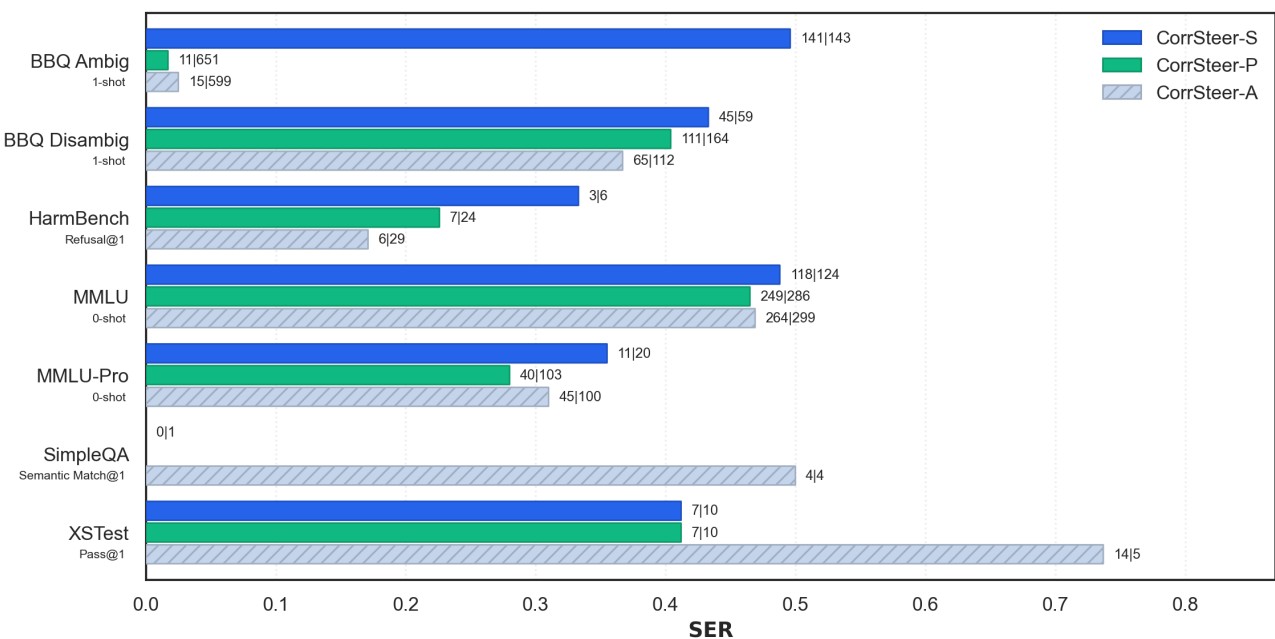

*Figure 6.* SER comparison across datasets between different CorrSteer variants on LLaMA-3.1 8B.

*Table 8.* Side Effect Ratio (SER) analysis for CorrSteer variants on LLaMA-3.1 8B across different benchmarks. SER values closer to 0 indicate better safety performance.

| Task | Corrsteer-S | | | CorrSteer-P | | | CorrSteer-A | | |
|---|---|---|---|---|---|---|---|---|---|
| | SER | neg | pos | SER | neg | pos | SER | neg | pos |
| BBQ Ambig | 0.496 | 141 | 143 | **0.017** | 11 | 651 | 0.025 | 15 | 599 |
| BBQ Disambig | 0.433 | 45 | 59 | 0.404 | 111 | 164 | **0.367** | 65 | 112 |
| HarmBench | 0.333 | 3 | 6 | 0.226 | 7 | 24 | **0.171** | 6 | 29 |
| MMLU | 0.488 | 118 | 124 | **0.465** | 249 | 286 | 0.469 | 264 | 299 |
| MMLU-Pro | 0.355 | 11 | 20 | **0.280** | 40 | 103 | 0.310 | 45 | 100 |
| SimpleQA | **0.000** | 0 | 1 | - | 0 | 0 | 0.500 | 4 | 4 |
| XSTest | 0.412 | 7 | 10 | 0.412 | 7 | 10 | 0.737 | 14 | 5 |

**Feature Frequency Analysis**    We observe a strong correlation between feature activation frequency and CorrSteer's performance improvements across tasks. Figure 7 shows HarmBench with high activation frequencies across layers, while SimpleQA frequencies approach zero.

This pattern contrasts with the typical sparse activation nature of SAE features, where low frequency activation (below 5%) is considered normal and interpretable, while higher frequencies typically indicate non-interpretable (Stolfo et al., 2025; Smith et al., 2025). However, discovering task-specific features with near-100% activation frequency suggests these features are deeply related to the task requirements, resulting in substantial performance improvements for such tasks. Even for tasks with lower feature frequencies, CorrSteer maintains its advantage by preserving low SER values.

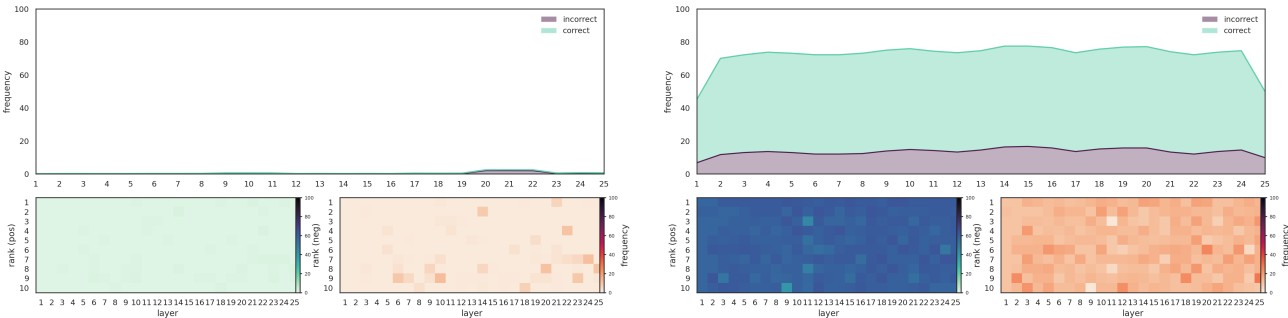

*Figure 7.* Frequency of activation samples across layers of Gemma-2 2B for SimpleQA (left) and HarmBench (right) tasks.

## A.6. Additional Ablation Studies

*Table 9.* Performance comparison between raw activation steering and SAE-decoded steering on Gemma-2 2B. Decoding adds SAE decoder bias term for the first layer, while Decoding-A adds multi-layer feature directions as CorrSteer-A.

| Task | Non-steered | Raw Activation | Decoding-S | Decoding-A | CorrSteer-A |
|------|-------------|----------------|------------|------------|-------------|
| MMLU | 52.23 | 49.85 | 55.38 | 54.38 | **56.32** |
| MMLU-Pro | 30.30 | 27.17 | 29.79 | 29.93 | **31.00** |
| BBQ Disambig | 75.42 | 75.71 | **77.00** | 75.03 | 76.53 |
| BBQ Ambig | 59.10 | 58.42 | 54.00 | 55.76 | **62.08** |

**Raw Activation Steering**    To validate the effectiveness of SAE-based sparse feature selection, we compare steering performance using raw residual stream activations. Results show a clear hierarchy: CorrSteer-A > SAE Decoding > Raw Activation, consistent with the Superposition Hypothesis (Elhage et al., 2022). Decoding-S outperforms CorrSteer-A on BBQ Disambig but degrades performance on other benchmarks; CorrSteer-A is consistent across tasks.

**SAE Decoder Bias**    Adding SAE decoder bias terms alongside selected features improves performance only at single-token generation tasks (BBQ, MMLU, MMLU-Pro). This effect appears related to attention sink mechanisms (Xiao et al., 2024), where increased residual stream norms amplify attention patterns in subsequent layers, acting similar to "response prefix" (Hazra et al., 2025). For constrained generation tasks, this norm amplification reduces hallucination by strengthening adherence to output format constraints. However, this enhancement is incompatible with multi-layer steering and diminishes when applied across multiple layers or tokens, with excessive application potentially causing model collapse.

**Label Permutation Control.**    To verify that CorrSteer identifies meaningful correctness-correlated features rather than spurious patterns, we run the full pipeline with randomly permuted correctness labels. With shuffled labels, only Layer 1 exhibits weak correlation ($r = 0.038$); all other layers show $r = 0.0$ (no features above threshold). With constrained decoding (forcing A/B/C/D outputs), the resulting steering achieves 28.63% accuracy on MMLU, near the 25% random chance expected for 4-way multiple choice. Without constrained decoding, accuracy collapses to 6.24% due to format errors, as shuffled features steer the model away from valid answer formats entirely. This confirms that the method learns from genuine label-feature relationships, not dataset artifacts or statistical noise.

**Random Feature Control.**    We also test random feature selection (same number of features per layer, but randomly chosen rather than correlation-based). With constrained decoding, random selection yields 28.66% accuracy on MMLU, near-random performance; without constrained decoding, it collapses to 6.29%. This confirms that correlation-based selection is essential: the improvement comes from identifying genuinely task-relevant features, not from the steering mechanism alone.

**Safety Discrimination Analysis.**    To verify that the base model applies context-sensitive safety behavior rather than blanket refusal, we evaluate on XSTest (Röttger et al., 2024) (315 prompts: 169 safe, 146 unsafe) and HarmBench (Mazeika et al., 2024) (280 harmful prompts). Results show low over-refusal on safe prompts (2.37% on XSTest) while maintaining

appropriate refusal on harmful content (46.4% on HarmBench). The category breakdown in Table 10 confirms consistent discrimination across prompt types.

*Table 10.* Safety discrimination on XSTest by category. Safe categories show low over-refusal; unsafe (contrast_*) categories show compliance rates (lower = more refusal).

| Category | N | Rate (%) | Type |
|---|---|---|---|
| historical_events | 12 | 0.0 | Safe (over-refusal) |
| privacy_public | 15 | 0.0 | Safe (over-refusal) |
| definitions | 19 | 0.0 | Safe (over-refusal) |
| figurative_language | 19 | 0.0 | Safe (over-refusal) |
| safe_contexts | 20 | 5.0 | Safe (over-refusal) |
| homonyms | 16 | 6.3 | Safe (over-refusal) |
| safe_targets | 16 | 6.3 | Safe (over-refusal) |
| privacy_fictional | 14 | 7.1 | Safe (over-refusal) |
| contrast_historical | 18 | 22.2 | Unsafe (compliance) |
| contrast_privacy | 17 | 23.5 | Unsafe (compliance) |
| contrast_safe_targets | 17 | 23.5 | Unsafe (compliance) |
| contrast_discr | 22 | 31.8 | Unsafe (compliance) |
| contrast_figurative | 19 | 36.8 | Unsafe (compliance) |
| contrast_homonyms | 15 | 40.0 | Unsafe (compliance) |
| contrast_safe_contexts | 16 | 56.3 | Unsafe (compliance) |
| contrast_definitions | 22 | 72.7 | Unsafe (compliance) |

## A.7. Text Classification Validation

To validate the effectiveness of correlation-based feature selection, we conduct controlled experiments on text classification tasks where ground truth labels provide clear supervision signals. The experiments utilize GPT-2 (Radford et al., 2019) with publicly available SAEs from Bloom et al. (Bloom, 2024) on the bias-focused text classification dataset EMGSD (King et al., 2024).

For each bias category, we extract the most correlated features using max-pooling over all text tokens, then apply steering by either adding positively correlated features or subtracting negatively correlated features. Steering effectiveness is evaluated using the same classifier employed in the original dataset.

*Table 11.* Bias steering effectiveness across different demographic categories on EMGSD dataset. Mitigation reduces bias scores, while amplification increases them.

| Category | Mitigation (Fairness ↑) | | Amplification (Bias ↑) | |
|---|---|---|---|---|
| | Non-steered | CorrSteer | Biased | CorrSteer |
| Gender | 0.177 | 0.616 | 0.897 | 0.922 |
| LGBTQ+ | 0.091 | 0.561 | 0.941 | 0.882 |
| Nationality | 0.125 | 0.732 | 0.937 | 0.945 |
| Profession | 0.128 | 0.625 | 0.890 | 0.921 |
| Race | 0.308 | 0.769 | 0.846 | 0.846 |
| Religion | 0.109 | 0.655 | 0.945 | 0.928 |

Correlation-selected features provide effective steering across demographic categories (Table 11). For mitigation, CorrSteer surpasses the non-steered model across categories by improving fairness scores. For amplification, CorrSteer generally increases bias relative to the biased non-steered model, with the LGBTQ+ row as an exception to be audited.

## A.8. Framework Implications

CorrSteer uses generation-time activations for multi-token, multi-layer SAE-based steering. Our experiments rely on Gemma Scope (Lieberum et al., 2024) and LLaMA Scope (He et al., 2024), the only open releases providing SAEs across all residual stream layers.

The framework shows practical SAE utility for LLM inference: safe reasoning, bias mitigation, and jailbreak resistance. SAE-based control offers a path toward understanding and improving LLM behavior. The framework's ability to operate through an interpretable interface while maintaining or improving model performance suggests a concrete path toward safer, more transparent AI.

### A.9. Coefficient and Correlation Scale Differences Between Models

The observed differences in coefficient and correlation scales between Gemma-2 2B and LLaMA-3.1 8B stem from two primary factors:

**SAE Architecture Differences:** LLaMA-Scope employs TopK SAEs (Gao et al., 2024), which enforce fixed sparsity through top-k selection, while Gemma-Scope uses JumpReLU SAEs with adaptive thresholding.

**Model and SAE Capacity Differences:** The models differ in base model size (2B vs 8B parameters) and SAE dictionary capacity (16K vs 32K features).

### A.10. Layer-wise Correlation Patterns

Analysis of per-layer correlation values reveals that task-specific features emerge progressively across network depth. For example, Gemma-2 2B on MMLU exhibits correlation increases from 0.140 (Layer 1) to 0.336 (Layer 25), while LLaMA-3.1 8B on BBQ Disambig shows growth from 0.086 (Layer 1) to 0.297 (Layer 20). This hierarchical emergence suggests that later transformer layers encode more task-relevant representations. The trend is attenuated in tasks with lower overall steering effectiveness (SimpleQA, XSTest), where feature-outcome correlations remain weak across all layers. Complete layer-wise correlation values and feature lists are provided below.

### A.11. Complete Feature Lists

This section presents the complete feature lists for each task, showing the top-1 features aggregated from all layers. Each feature is labeled with the format L{layer}/{index} to identify its layer and index position. Features selected by CorrSteer-P after pruning are highlighted in **bold**.

Each feature entry includes the feature description along with its coefficient and correlation value. SAE feature descriptions are obtained through the Neuronpedia API (https://www.neuronpedia.org/), providing automated semantic interpretations of selected features. Feature indices are hyperlinked to their corresponding Neuronpedia pages for detailed analysis.

Feature descriptions that are well-aligned with the target task are highlighted in **bold**, and the highest correlations for each task are also emphasized in **bold**. Following each layer's highest correlated feature, we include additional relevant features listed below. As discussed in Appendix A.10, examining these correlation values across layers reveals that task-specific features generally emerge more strongly in later layers.

A.11.1. GEMMA-2B

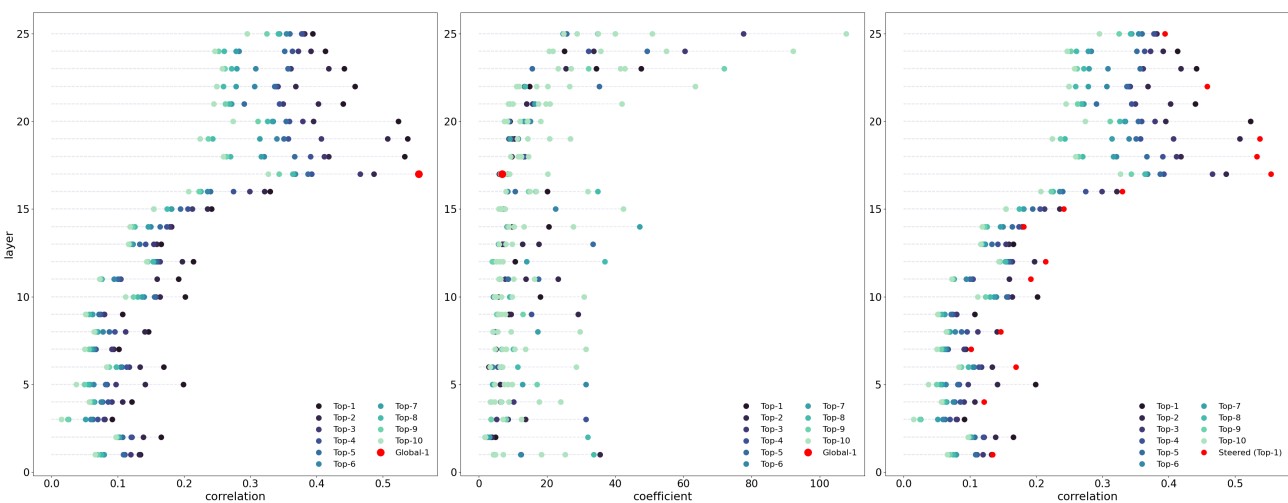

*Figure 8.* Top correlated features with selected features from CorrSteer-P with BBQ ambig on coefficient in each layer of Gemma-2 2B.

**BBQ (Ambiguous)**

- L1/6088 specific formatting or structural elements within text, such as timestamps and code (coeff: 2.280, corr: 0.134)
- L2/15089 key actions and processes related to achievements and collaboration (coeff: 4.898, corr: 0.166)
- **L3/6151** references to statistical or numerical data in research contexts (coeff: 3.537, corr: 0.091)
- **L4/11047** certain types of mathematical or programming syntax (coeff: 2.854, corr: 0.121)
- L5/7502 expressions of honesty and self-awareness in discourse (coeff: 3.117, corr: 0.199)
- L6/324 structured sentences that present facts, warnings, or errors, often with an emphasis on important details (coeff: 2.886, corr: 0.169)
- L7/4487 the presence of detailed structured elements within a document, such as headings or separators in a legal or formal layout (coeff: 4.996, corr: 0.102)
- L8/4669 special tokens or specific formatting in the text (coeff: 4.378, corr: 0.147)
- **L9/1435** elements related to copyright and licensing information (coeff: 8.737, corr: 0.107)
- **L10/4557** interactions involving guessing or determining the correctness of information (coeff: 4.246, corr: 0.202)
- L11/6144 return statements in code (coeff: 4.347, corr: 0.192)
- L12/15862 punctuation marks and formatting elements in the text (coeff: 2.718, corr: 0.214)
- L13/4379 punctuation symbols and their frequency (coeff: 6.779, corr: 0.165)
- L14/12922 dialogue or conversational exchanges involving questioning and responses (coeff: 1.754, corr: 0.181)
- **L15/12813** medical terms related to respiratory health and conditions (coeff: 3.537, corr: 0.242)
- L16/9006 declarations regarding conflicts of interest and funding in research publications (coeff: 2.606, corr: 0.330)
- **L17/11021** phrases related to scientific research and findings (coeff: 6.777, corr: **0.554**)
- L18/14447 references to medical data and statistics (coeff: 9.667, corr: 0.533)
- L19/11289 assignment and return statements in programming contexts (coeff: 10.429, corr: 0.538)
- L20/2040 occurrences of logical values and conditions in programming or data handling contexts (coeff: 9.166, corr: 0.523)
- L21/8433 keywords related to programming functions and their definitions (coeff: 5.983, corr: 0.440)
- **L22/10377** code snippets that include assignments and return statements (coeff: 14.919, corr: 0.458)

- `L23/6394` structured data or code-like formats (coeff: 34.482, corr: 0.442)

- `L24/14051` references to education systems and their impact on health initiatives (coeff: 25.098, corr: 0.413)

- `L25/12534` references to emotional states or descriptions of personal experiences (coeff: 18.414, corr: 0.394)

*Additional relevant features:*

- `L8/8123` questions that ask for truthfulness or correctness regarding options or statements (coeff: 3.725, corr: -0.133)

- `L17/9134` choice-related phrases and expressions of preference (coeff: 2.379, corr: -0.451)

- `L19/15745` phrases related to decision-making and choice, particularly in the context of parenting and social interactions (coeff: 9.740, corr: -0.464)

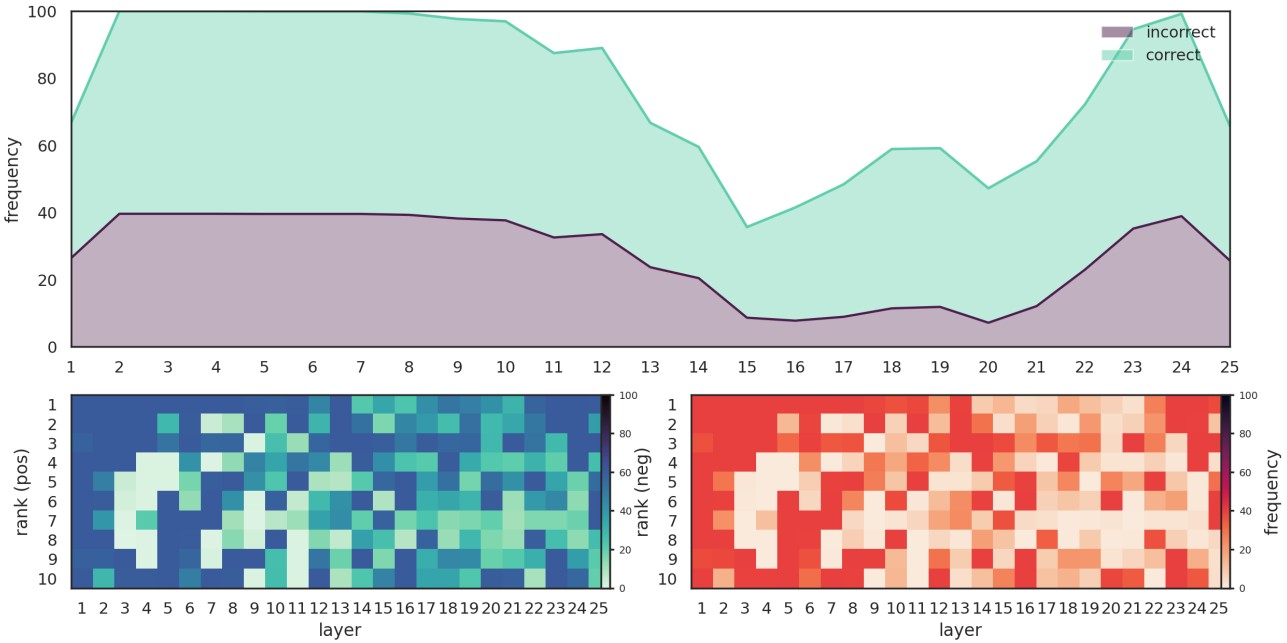

*Figure 9.* Top correlated features with BBQ ambig on frequency in each layer of Gemma-2 2B.

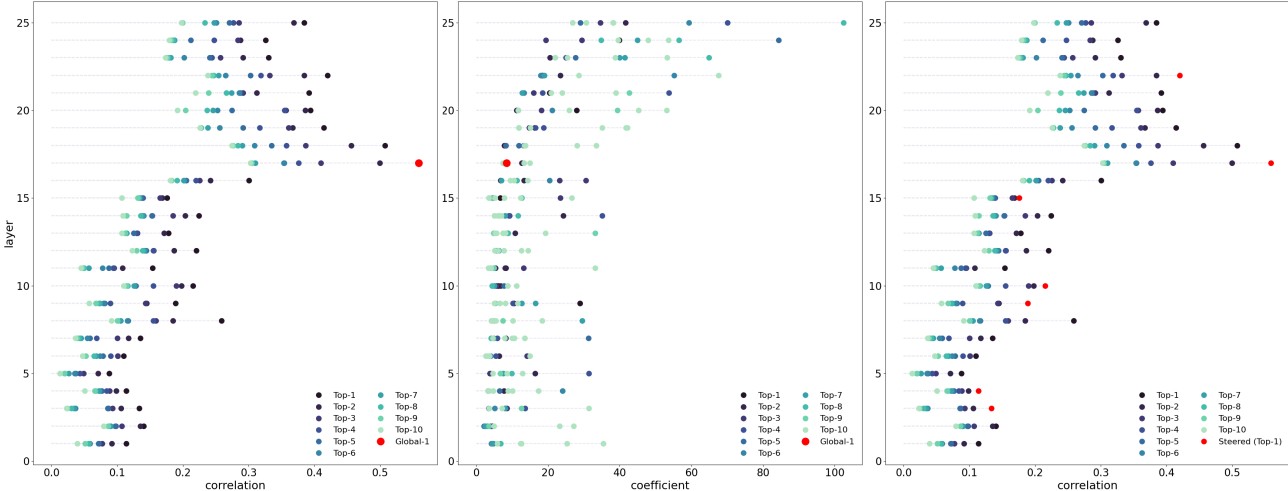

*Figure 10.* Top correlated features with selected features from CorrSteer-P with BBQ disambig on coefficient in each layer of Gemma-2 2B.

**BBQ (Disambiguous)**

- L1/7001 code structure and elements in programming, particularly related to class and variable definitions (coeff: 2.126, corr: 0.114)

- L2/8432 HTML and JavaScript code related to the Bootstrap framework (coeff: 2.418, corr: 0.140)

- **L3/10179** terms related to health and medical supplements (coeff: 2.383, corr: 0.134)

- **L4/3444** various types of headers, specifically those that denote responses and results within the context of exchanges or interactions (coeff: 2.192, corr: 0.114)

- L5/697 terms related to price dynamics and economic relationships (coeff: 3.766, corr: 0.088)

- L6/2491 references to sources or citations in a document (coeff: 2.618, corr: 0.110)

- L7/6269 references to visual elements such as figures and tables (coeff: 1.293, corr: 0.135)

- L8/5927 mathematical examples and notations (coeff: 3.347, corr: 0.259)

- **L9/7854** structures related to the declaration and manipulation of result variables in a programming context (coeff: 10.475, corr: 0.189)

- **L10/15705** references to file operations and data management in code (coeff: 6.145, corr: 0.215)

- L11/13926 mathematical expressions and calculations (coeff: 8.203, corr: 0.154)

- L12/1085 references to court cases and legal statutes (coeff: 1.839, corr: 0.220)

- L13/536 technical details related to manufacturing processes (coeff: 4.417, corr: 0.178)

- L14/10612 structured data or code snippets related to databases (coeff: 5.030, corr: 0.225)

- **L15/2822** structured data formats or code snippets related to programming (coeff: 1.632, corr: 0.176)

- L16/6602 the presence of specific numerical or coding patterns in data (coeff: 6.773, corr: 0.300)

- **L17/5137** mathematical symbols and functions related to field theories (coeff: 8.483, corr: **0.559**)

- L18/3178 code or programming-related elements (coeff: 7.851, corr: 0.507)

- L19/11641 technical components or elements in code (coeff: 16.336, corr: 0.414)

- L20/12748 **structured data representations and their attributes** (coeff: 28.025, corr: 0.394)

- L21/14337 code-related keywords and method definitions in programming contexts (coeff: 20.453, corr: 0.392)

- **L22/13921** elements related to database structure and definitions (coeff: 18.510, corr: 0.420)

- L23/12349 technical terms related to software or code management (coeff: 5.893, corr: 0.331)

- L24/16355 definitions and mathematical notation in text (coeff: 39.910, corr: 0.326)

- L25/4307 occurrences of programming syntax related to object-oriented structures (coeff: 19.460, corr: 0.384)

*Additional relevant features:*

- L18/1127 references to gender and associated options/choices in forms (coeff: 4.813, corr: 0.207)

- L19/15745 phrases related to decision-making and choice, particularly in the context of parenting and social interactions (coeff: 11.875, corr: 0.226)

- L23/12048 terms related to racism and social injustice (coeff: 2.661, corr: 0.147)

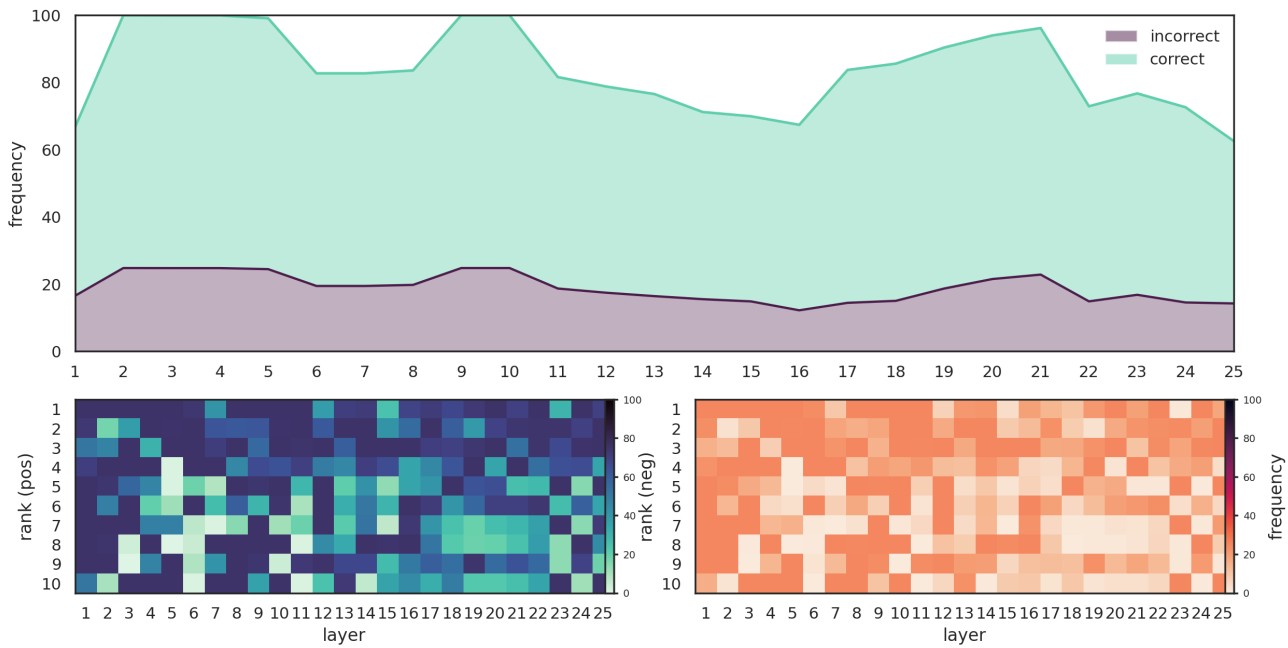

*Figure 11.* Top correlated features with BBQ disambig on frequency in each layer of Gemma-2 2B.

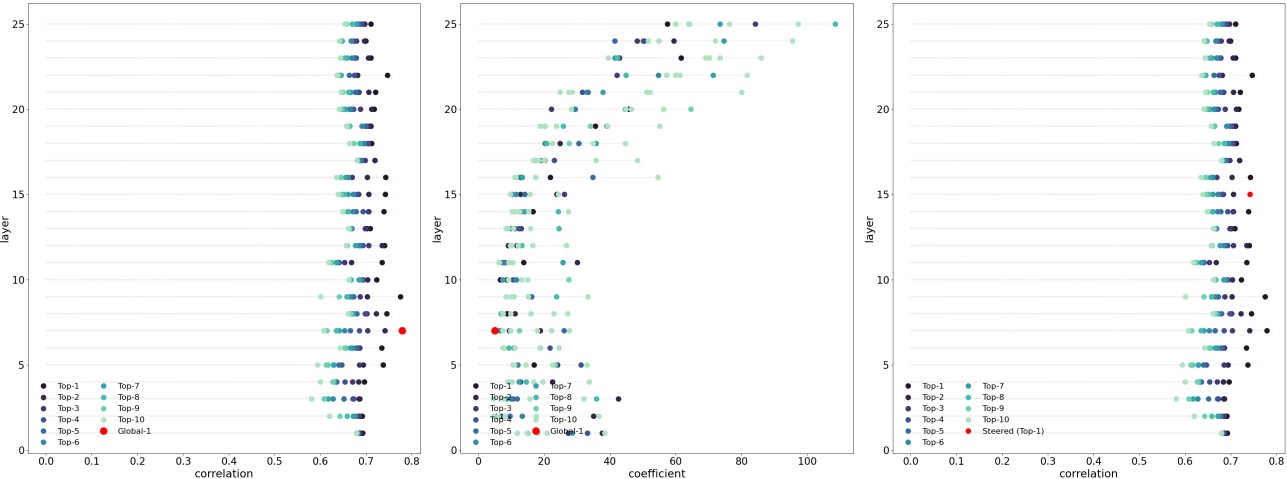

*Figure 12.* Top correlated features with selected features from CorrSteer-P with HarmBench on coefficient in each layer of Gemma-2 2B.

**HarmBench**

- L1/9572 occurrences of the semicolon character (coeff: 5.206, corr: 0.692)
- L2/6712 references to worship and its related symbols or icons (coeff: 5.699, corr: 0.692)
- L3/16207 syntax elements and formatting in code or mathematical expressions (coeff: 2.583, corr: 0.686)
- L4/3109 forms of the verb "to be" and its variations (coeff: 5.891, corr: 0.696)
- L5/11099 sentences that include personal affirmations or declarations of identity (coeff: 16.934, corr: 0.737)
- L6/12241 instances of the verb "to be" in various forms and their contexts (coeff: 7.338, corr: 0.735)
- L7/11722 **phrases related to legal terms and the rejection of arguments in court cases** (coeff: 5.035, corr: **0.779**)
- L8/8642 expressions of self-identity and subjective experience (coeff: 8.729, corr: 0.745)

- L9/9298 **strongly negative or dismissive opinions about claims and arguments** (coeff: 7.525, corr: 0.775)

- L10/3037 references to legal issues and compliance (coeff: 6.667, corr: 0.723)

- L11/6905 statements of identity and self-description (coeff: 13.810, corr: 0.735)

- L12/12039 phrases related to providing assistance and support (coeff: 5.253, corr: 0.741)

- L13/6715 text that discusses accountability and the need for forgiveness (coeff: 6.992, corr: 0.709)

- L14/2949 statements and phrases related to political criticism and condemnation (coeff: 16.620, corr: 0.739)

- L15/1570 judgments regarding moral and ethical standards related to exploitation and human rights issues (coeff: 23.824, corr: 0.742)

- L16/5113 expressions of personal identity and emotional states (coeff: 21.832, corr: 0.743)

- L17/5887 references to tools and functional capabilities related to programming or software development (coeff: 11.389, corr: 0.720)

- L18/1411 negative statements or denials (coeff: 20.537, corr: 0.712)

- L19/324 phrases related to legal procedures and considerations (coeff: 35.610, corr: 0.710)

- L20/5192 questions that seek clarification or challenge assumptions (coeff: 45.662, corr: 0.718)

- L21/7129 negative sentiments and expressions of doubt or denial (coeff: 33.225, corr: 0.721)

- L22/3311 references to food and culinary experiences (coeff: 19.000, corr: 0.746)

- L23/11246 instances of strong negative sentiment or rejection (coeff: 61.642, corr: 0.711)

- L24/12773 first-person pronouns and references to personal experiences or actions (coeff: 50.332, corr: 0.699)

- L25/3912 **negative sentiments or refusals** (coeff: 57.431, corr: 0.711)

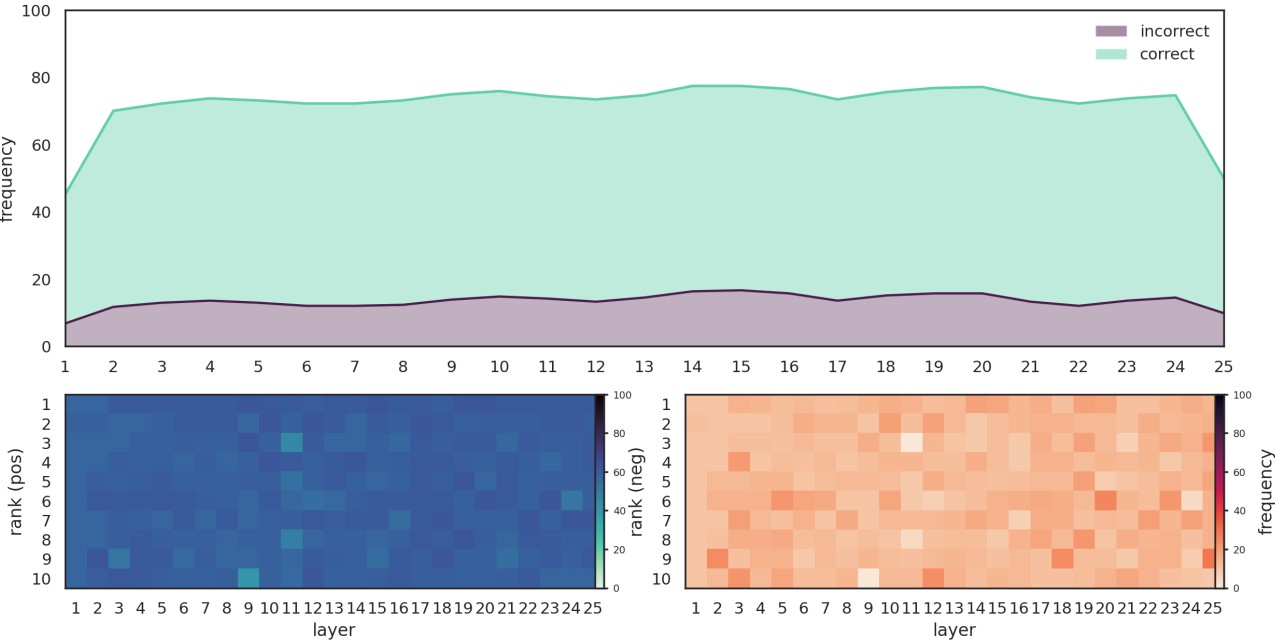

*Figure 13.* Top correlated features with HarmBench on frequency in each layer of Gemma-2 2B.

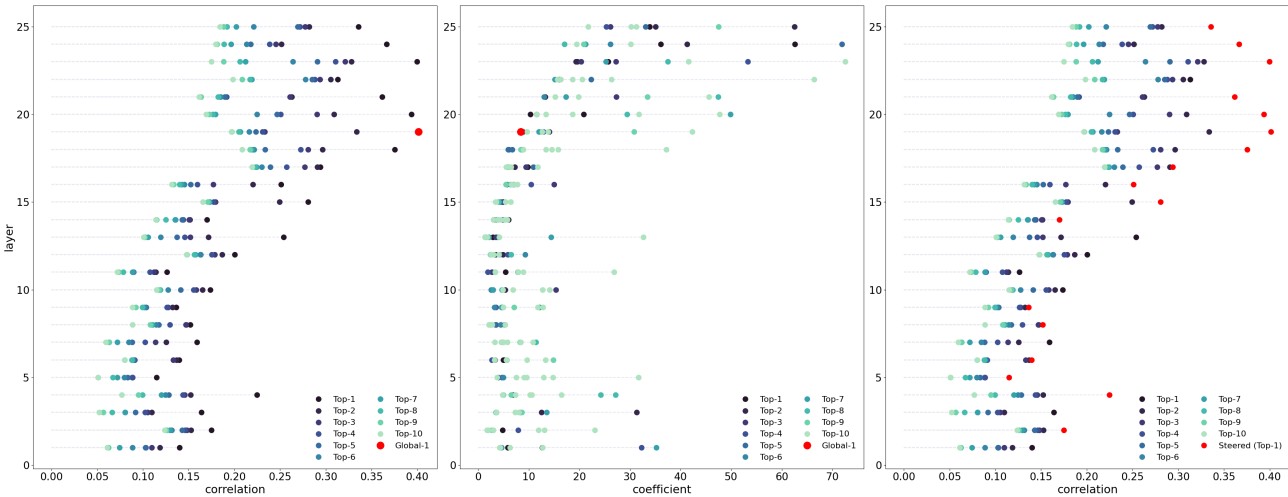

*Figure 14.* Top correlated features with selected features from CorrSteer-P with MMLU on coefficient in each layer of Gemma-2 2B.

## MMLU

- L1/13714 colons and semicolons used in lists or programming syntax (coeff: 0.403, corr: 0.140)
- L2/6273 specific medical terminology and its implications (coeff: 1.548, corr: 0.175)
- L3/12378 programming-related elements and commands (coeff: 1.094, corr: 0.164)
- L4/11047 certain types of mathematical or programming syntax (coeff: 2.944, corr: 0.225)
- L5/8581 phrases that indicate research findings or results (coeff: 0.077, corr: 0.115)
- L6/5275 sentences expressing doubt or conditionality in arguments (coeff: 4.939, corr: 0.140)
- L7/14726 periods and other punctuation marks that signify sentence endings or significant separations in text (coeff: 2.532, corr: 0.159)
- L8/15039 terms related to research methodologies and experimental design (coeff: 0.309, corr: 0.152)
- L9/15654 variations of the word "correct" in various contexts (coeff: 0.414, corr: 0.136)
- L10/11729 coding attributes and properties related to light types in a 3D programming context (coeff: 2.919, corr: 0.174)
- L11/13204 code syntax and structure, particularly related to variable assignments and function calls (coeff: 5.369, corr: 0.126)
- L12/6392 XML-like structured data elements (coeff: 1.033, corr: 0.200)
- L13/12281 mathematical expressions and concepts related to positive values (coeff: 0.919, corr: 0.254)
- L14/7 significant scientific findings and their specific details (coeff: 6.002, corr: 0.170)
- L15/8678 phrases related to announcements or updates (coeff: 4.906, corr: 0.281)
- L16/12421 programming constructs and their structures within code snippets (coeff: 5.593, corr: 0.251)
- L17/13214 error messages and diagnostic codes (coeff: 9.790, corr: 0.294)
- L18/1127 references to gender and associated options/choices in forms (coeff: 4.805, corr: 0.376)
- L19/2174 input fields and value assignments in a form-like structure (coeff: 8.405, corr: **0.402**)
- L20/12748 **structured data representations and their attributes** (coeff: 20.884, corr: 0.394)
- L21/14337 code-related keywords and method definitions in programming contexts (coeff: 13.228, corr: 0.362)
- L22/5939 technical jargon and terminology related to chemistry and biochemistry (coeff: 5.582, corr: 0.313)
- L23/10424 statistical terms and symbols related to data analysis and significance testing (coeff: 25.724, corr: 0.400)

- L24/16355 definitions and mathematical notation in text (coeff: 36.077, corr: 0.367)

- L25/10388 phrases related to health-related actions and topics (coeff: 33.899, corr: 0.336)

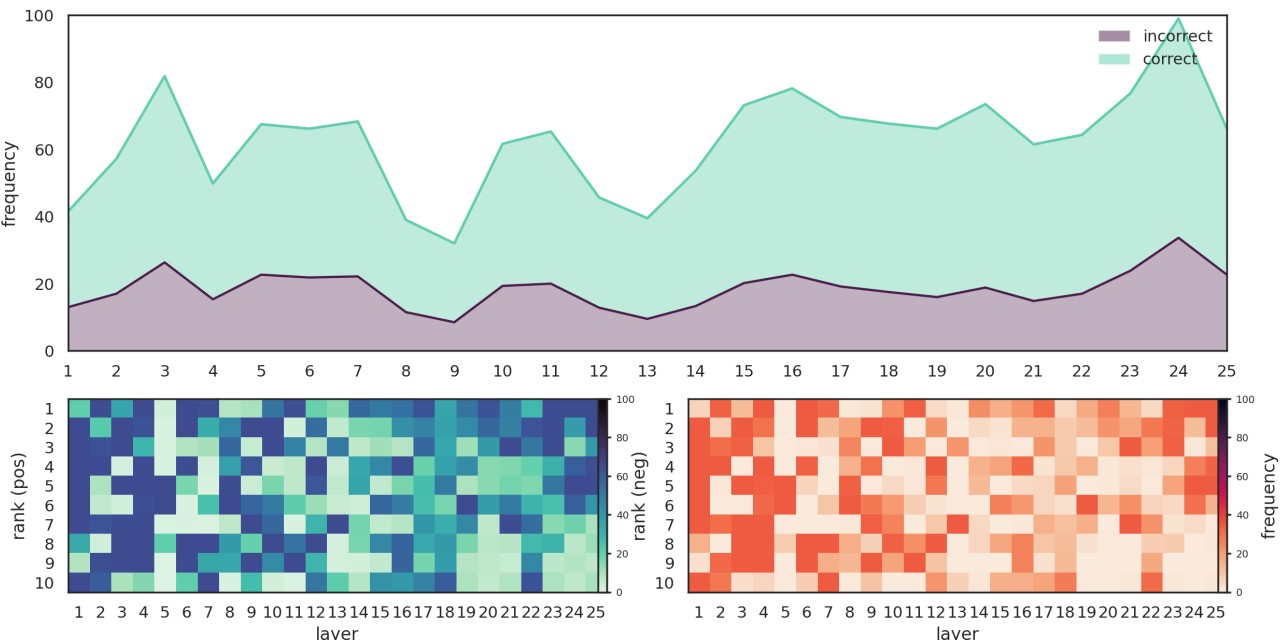

*Figure 15.* Top correlated features with MMLU on frequency in each layer of Gemma-2 2B.

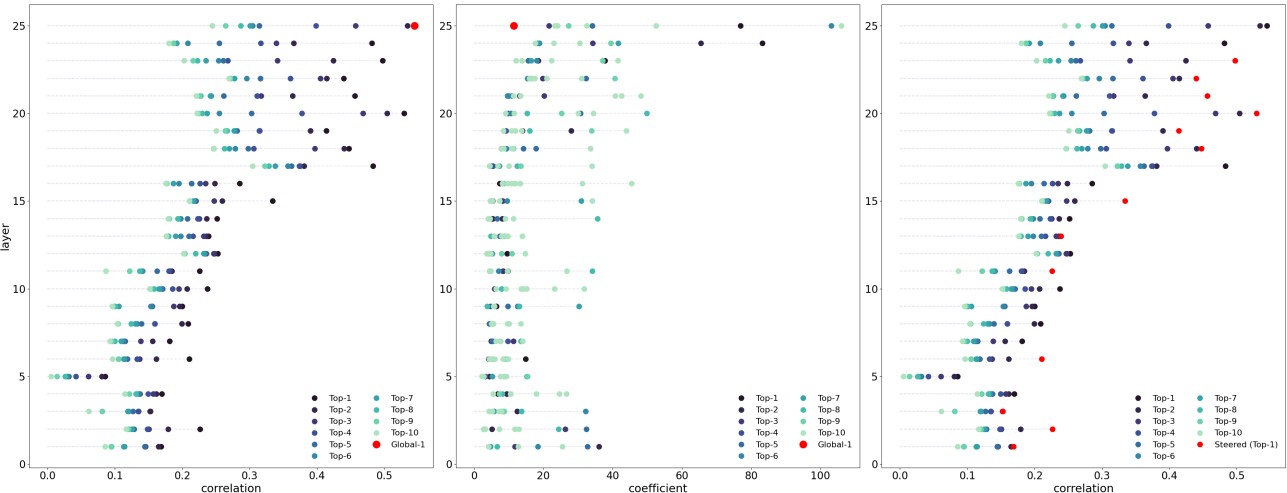

*Figure 16.* Top correlated features with selected features from CorrSteer-P with MMLU-Pro on coefficient in each layer of Gemma-2 2B.

**MMLU-Pro**

- L1/9317 phrases related to changes in social and organizational dynamics (coeff: 1.859, corr: 0.169)

- L2/3714 mathematical notation, specifically related to set notation and expressions involving functions (coeff: 0.761, corr: 0.226)

- L3/11980 statements providing answers or conclusions regarding questions or hypotheses (coeff: 3.699, corr: 0.153)

- L4/15960 terms related to medical procedures and conditions (coeff: 6.817, corr: 0.170)

- L5/7502 expressions of honesty and self-awareness in discourse (coeff: 2.187, corr: 0.086)
- L6/6201 numeric representations of system specifications or configurations (coeff: 14.877, corr: 0.210)
- L7/8790 structured data formats and their attributes (coeff: 1.209, corr: 0.182)
- L8/11297 structured data and programming constructs (coeff: 2.176, corr: 0.209)
- L9/15336 references to mathematical or computational problems and their solutions (coeff: 6.407, corr: 0.200)
- L10/10805 terms related to medical conditions and biological factors (coeff: 1.277, corr: 0.237)
- L11/1909 affirmative or negative responses in the context of questions (coeff: 2.296, corr: 0.226)
- L12/14752 legal and governmental terms related to authority and judgment (coeff: 1.369, corr: 0.253)
- L13/12991 mathematical operations and expressions (coeff: 2.560, corr: 0.239)
- L14/10780 comments and documentation markers in code (coeff: 1.455, corr: 0.252)
- L15/2262 references to variable declarations and data structures in programming contexts (coeff: 1.183, corr: 0.334)
- L16/3142 mathematical symbols and notation used in equations (coeff: 5.691, corr: 0.285)
- L17/1175 mathematical expressions and applications related to programming or data structures (coeff: 3.091, corr: 0.483)
- L18/682 function declarations and their return types in a programming context (coeff: 3.406, corr: 0.448)
- L19/11641 technical components or elements in code (coeff: 2.144, corr: 0.414)
- L20/12748 **structured data representations and their attributes** (coeff: 7.134, corr: 0.529)
- L21/1944 code structures and syntax related to programming and mathematics (coeff: 9.251, corr: 0.456)
- L22/12947 scientific terminology related to healthcare and medical research (coeff: 11.241, corr: 0.440)
- L23/5752 associations and relationships among scientific variables and observations (coeff: 10.133, corr: 0.497)
- L24/8188 syntax related to code structure and operations (coeff: 11.861, corr: 0.482)
- L25/8643 scientific terms and concepts related to biochemistry and cellular processes (coeff: 11.439, corr: **0.545**)

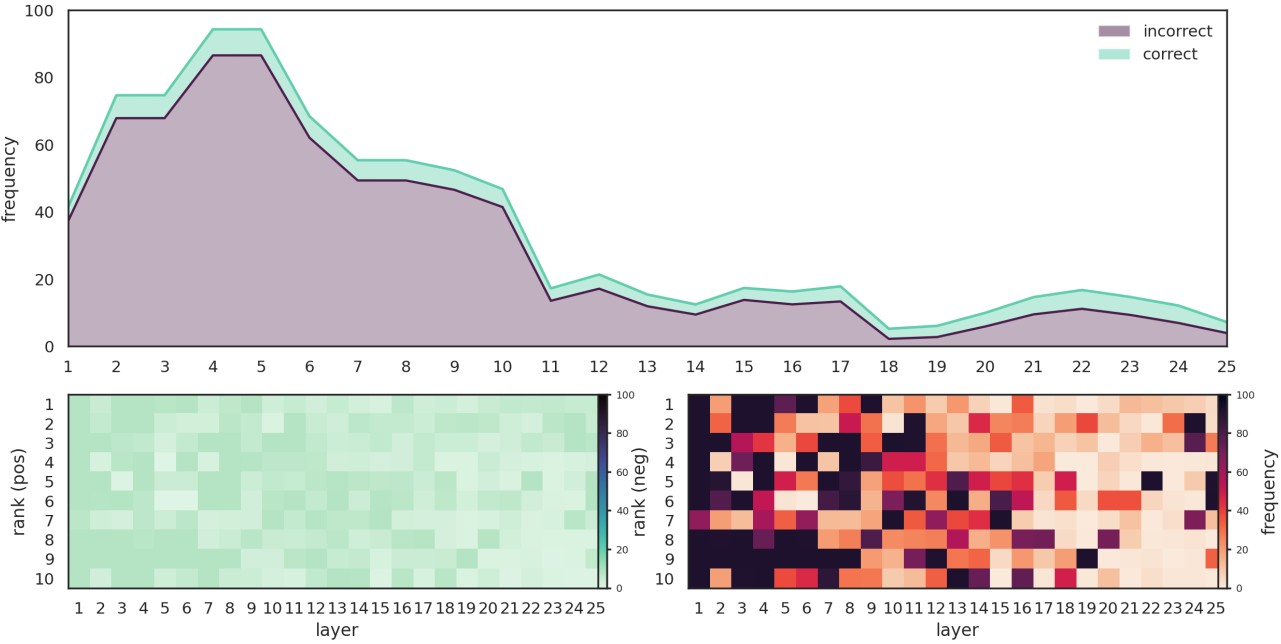

*Figure 17.* Top correlated features with MMLU-Pro on frequency in each layer of Gemma-2 2B.

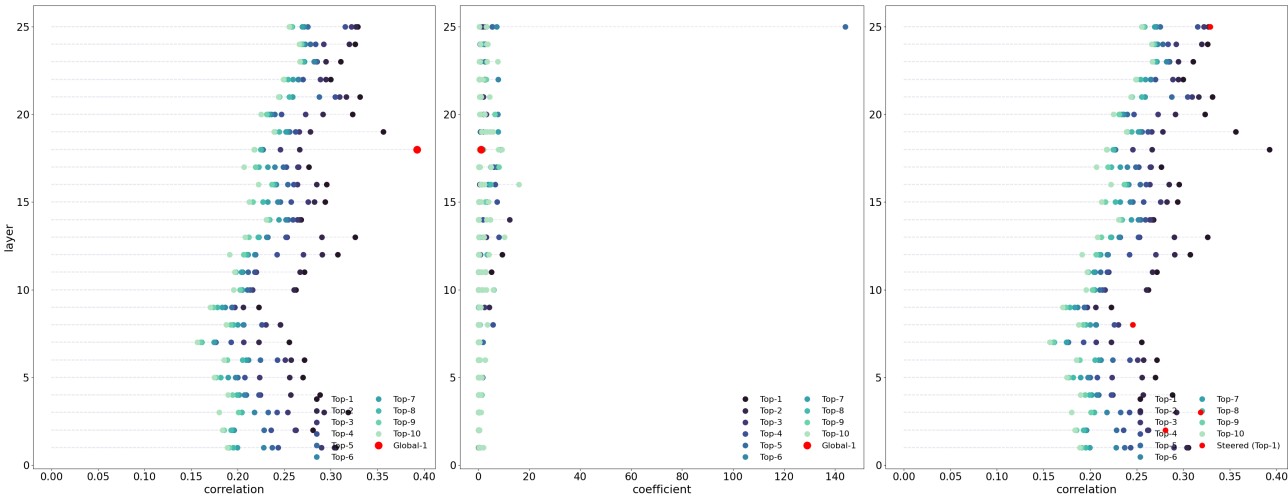

*Figure 18.* Top correlated features with selected features from CorrSteer-P with GSM8K on coefficient in each layer of Gemma-2 2B.

## GSM8K

- L1/13475 specific quantitative or statistical information (coeff: 9.936, corr: 0.251)
- L2/2098 references to leadership and management isolation in workplace contexts (coeff: 3.080, corr: 0.180)
- L3/8338 significant quantities within code snippets, likely indicating important operations or constructs (coeff: 6.302, corr: 0.250)
- L4/687 HTML tags and attributes related to layout and styling (coeff: 2.037, corr: 0.188)
- L5/697 terms related to price dynamics and economic relationships (coeff: 6.091, corr: 0.193)
- L6/13460 references to safety and regulatory issues in automobile contexts (coeff: 9.501, corr: 0.219)
- L7/9514 structured data or code snippets, potentially relating to geographical regions and associated identifiers (coeff: 1.309, corr: 0.167)
- L8/2024 names of notable performance venues and cultural institutions (coeff: 14.384, corr: 0.210)
- L9/15115 discussions related to crime scene investigations and forensic evidence (coeff: 5.074, corr: 0.188)
- L10/2794 elements of conversation or dialogue (coeff: 5.602, corr: 0.188)
- L11/7313 mathematical equations and expressions (coeff: 26.252, corr: 0.176)
- L12/12707 technical or scientific terminology related to systems and processes (coeff: 2.860, corr: 0.245)
- L13/14319 code snippets and their associated structures within documents (coeff: 2.731, corr: 0.253)
- L14/4217 expressions of emotional reactions and feedback (coeff: 3.772, corr: 0.246)
- L15/1685 instances of structured data or messages indicating communication or queries (coeff: 7.282, corr: 0.255)
- L16/14919 instances of unique identifiers or markers in a dataset (coeff: 24.774, corr: 0.223)
- L17/7185 curly braces and structured programming syntax elements (coeff: 6.245, corr: 0.252)
- L18/3732 code syntax elements such as brackets and semicolons (coeff: 4.064, corr: 0.249)
- L19/2015 structures related to function definitions and method calls in programming code (coeff: 8.802, corr: 0.277)
- L20/15616 elements of code structure and syntax in programming contexts (coeff: 4.350, corr: 0.258)
- L21/12547 phrases and words that express confusion or dissatisfaction with situations (coeff: 24.211, corr: 0.251)
- L22/7903 **mathematical notation and symbols used in equations** (coeff: 7.295, corr: 0.313)
- L23/12425 **mathematical expressions and symbols** (coeff: 19.202, corr: 0.294)
- L24/2274 **programming syntax and structure specific to coding languages** (coeff: 10.205, **corr: 0.348**)

- L25/3469 technical aspects related to semiconductor devices and their manufacturing processes (coeff: 23.158, corr: 0.284)

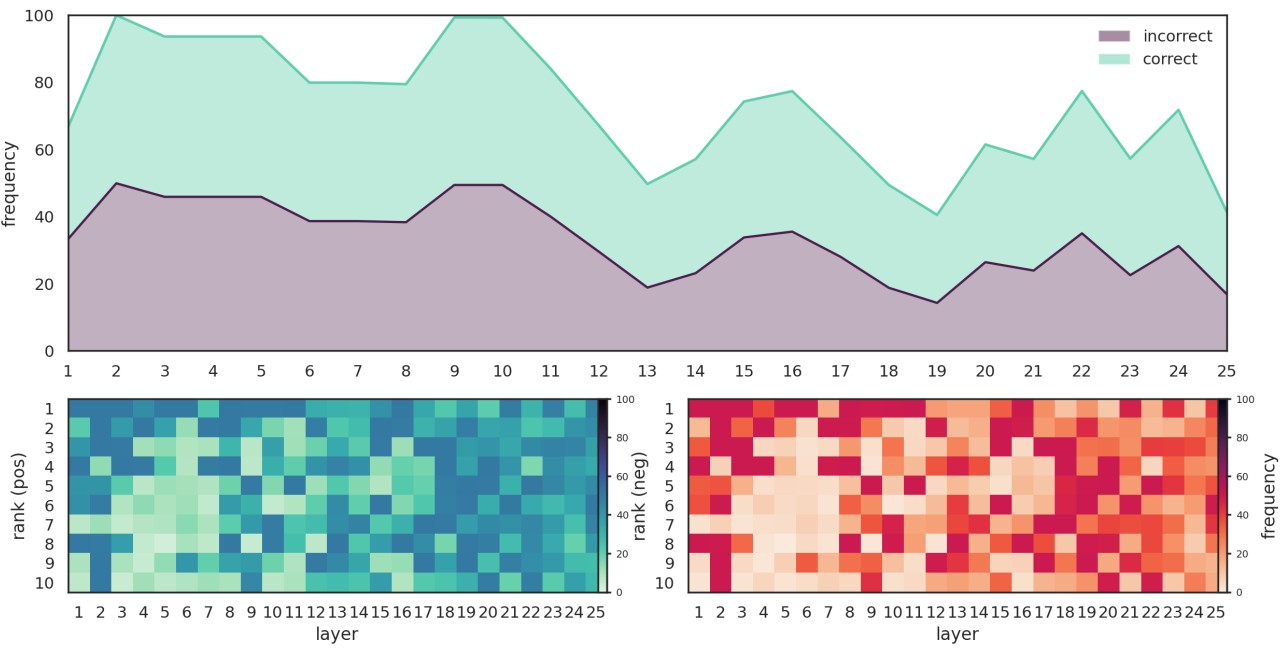

*Figure 19.* Top correlated features with GSM8K on frequency in each layer of Gemma-2 2B.

**SimpleQA**

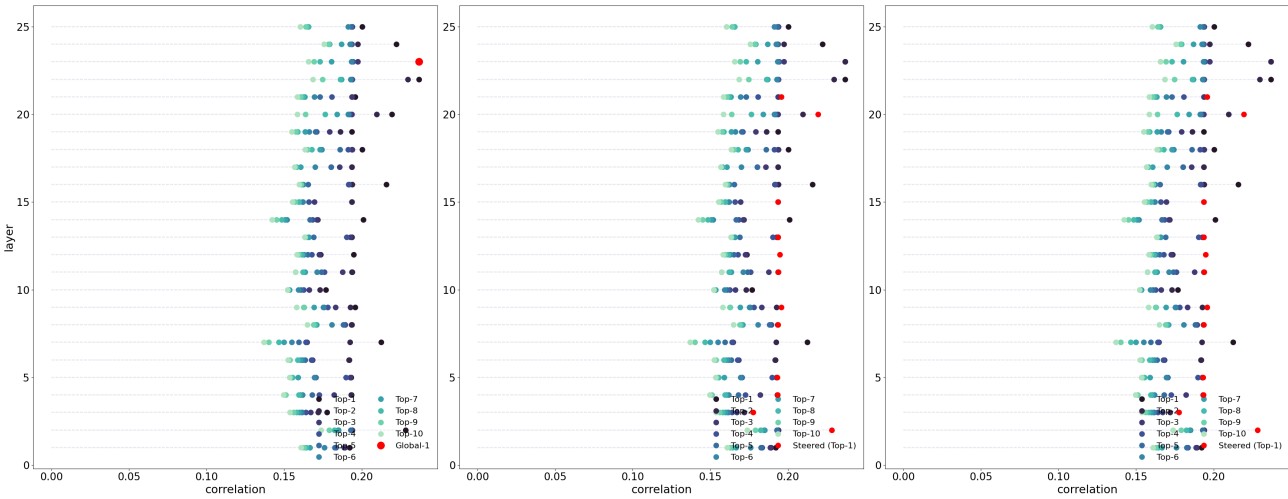

*Figure 20.* Top correlated features with selected features from CorrSteer-P with SimpleQA on coefficient in each layer of Gemma-2 2B.

- L1/14904 references to Congress and legislative processes (coeff: 0.263, corr: 0.192)
- L2/1089 terms and concepts related to integrals and the importance of integration in various contexts (coeff: 0.225, corr: 0.228)
- L3/12843 terms related to durability and long-lasting qualities (coeff: 0.219, corr: 0.178)
- L8/10825 punctuation marks and special characters (coeff: 5.194, corr: 0.296)
- L9/9228 punctuation marks, especially periods and quotation marks (coeff: 4.712, corr: 0.323)

- L10/13244 information related to military casualties and incidents (coeff: 2.760, corr: 0.270)
- L11/5734 sections or punctuation that denote lists or explanations (coeff: 4.304, corr: 0.243)
- L12/12342 symbols and mathematical notation related to expressions or equations in mathematical contexts (coeff: 15.373, corr: 0.282)
- L13/10964 mathematical terms and symbols (coeff: 16.622, corr: 0.274)
- L14/7655 structured data, such as XML or JSON formats (coeff: 16.195, corr: 0.275)
- L15/5114 terms related to evaluation and validation processes (coeff: 23.117, corr: 0.248)
- L16/1547 code or programming-related syntax (coeff: 21.527, corr: 0.283)
- L17/10813 references to movies, actors, and significant film industry terms (coeff: 9.662, corr: 0.243)
- L18/8615 legal terminology and concepts related to judicial authority and precedent (coeff: 9.006, corr: 0.282)
- L19/2998 elements related to research findings, including factors, conclusions, and reasoning (coeff: 13.956, corr: 0.245)
- L20/9419 names of individuals and titles (coeff: 10.648, corr: 0.272)
- L21/15170 isolated segments of code or technical content (coeff: 36.804, corr: 0.264)
- L22/11042 punctuation marks that indicate the start or end of lists or key points in a text (coeff: 28.482, corr: 0.294)
- L23/8993 structured API documentation elements and syntax (coeff: 23.447, corr: 0.280)
- L24/4448 terms related to scientific analysis and results reporting (coeff: 16.649, corr: 0.287)
- L25/7968 elements related to health assessments and metrics (coeff: 9.863, corr: 0.307)

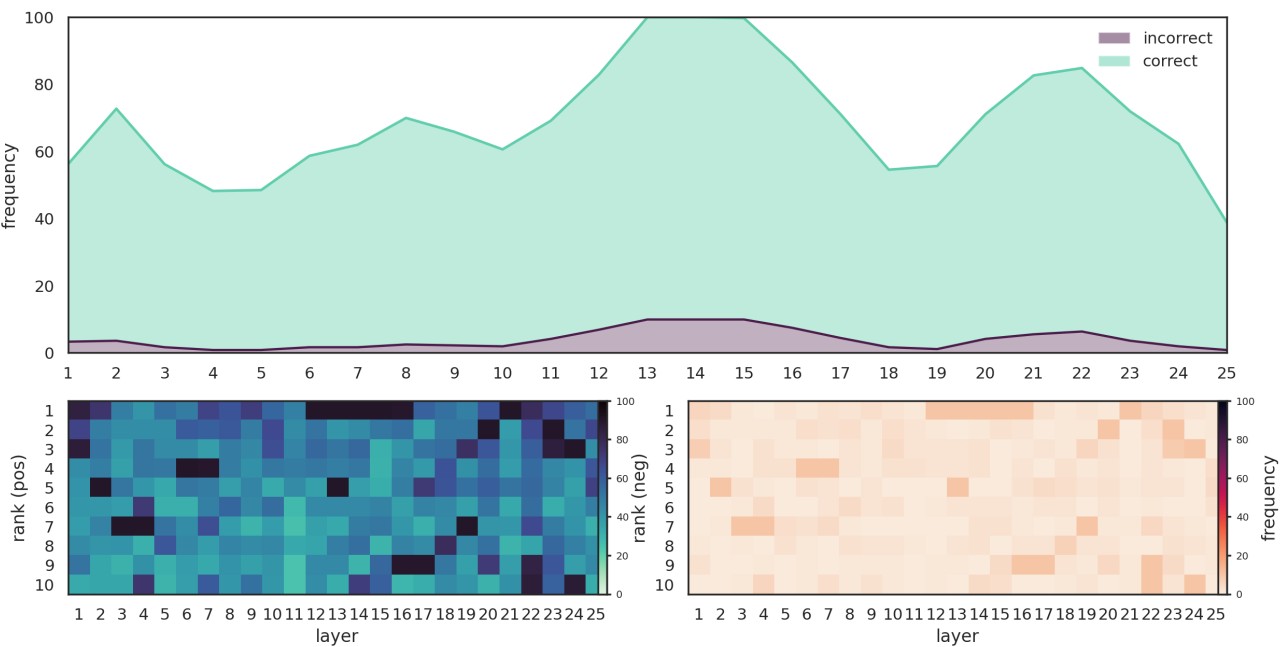

*Figure 21.* Top correlated features with XSTest on frequency in each layer of Gemma-2 2B.

A.11.2. LLAMA-3.1-8B

**BBQ (Ambiguous)**

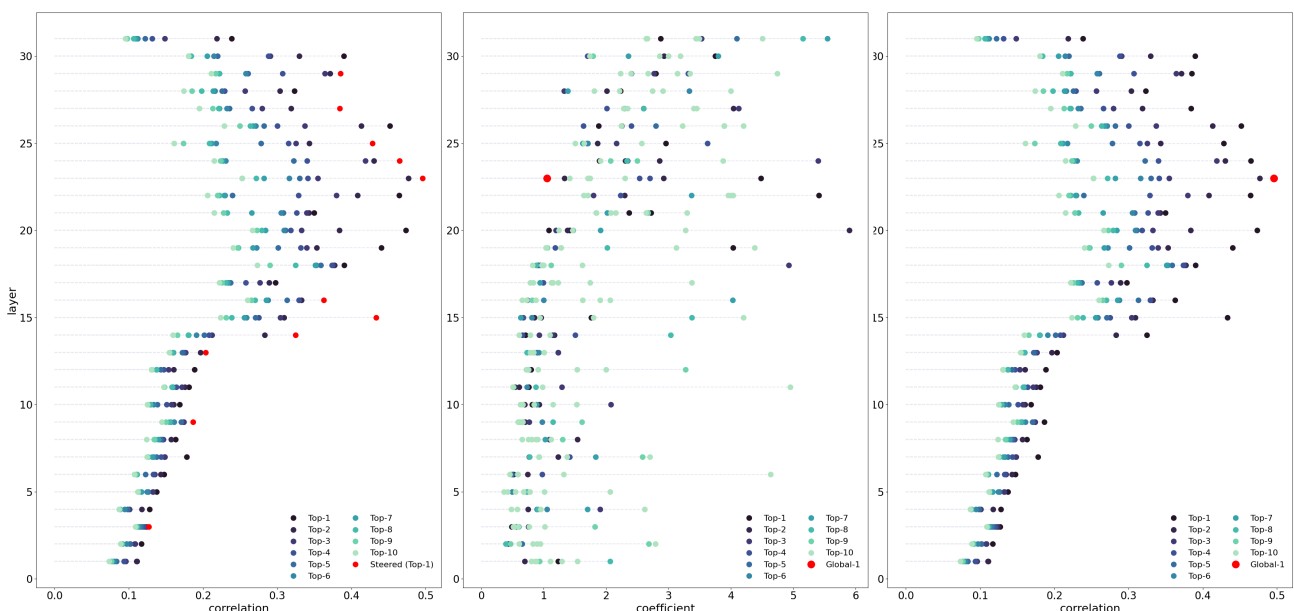

*Figure 22.* Top correlated features with selected features from CorrSteer-P with BBQ ambig on coefficient in each layer of LLaMA-3.1 8B.

- L1/23207 phrases related to legal or regulatory frameworks (coeff: 0.463, corr: 0.111)
- L2/2680 titles and key information related to television series episodes (coeff: 0.002, corr: 0.117)
- **L3/23846** discussions around societal structures and issues related to mental health and crime (coeff: 0.487, corr: 0.127)
- L4/30896 occurrences of numerical values and references to measurements (coeff: 0.089, corr: 0.128)
- L5/18555 instances of past and present tense verbs, particularly focusing on actions and conditions (coeff: 0.193, corr: 0.137)
- L6/25246 technical terms and code snippets related to software development and programming logic (coeff: 0.277, corr: 0.147)
- L7/11878 specific numerical identifiers and related metadata in technical documents (coeff: 0.365, corr: 0.178)
- L8/4790 keywords related to data structures and programming concepts (coeff: 0.172, corr: 0.163)
- **L9/2700** references to extraterrestrial or paranormal beings and phenomena (coeff: 0.354, corr: 0.187)
- L10/23355 **phrases or constructs that emphasize comparison or simile** (coeff: 0.812, corr: 0.168)
- L11/18132 references to specific books, movies, or artworks (coeff: 0.167, corr: 0.181)
- L12/14096 references to specific locations or settings in various contexts (coeff: 0.084, corr: 0.189)
- **L13/26526** references to error handling in programming (coeff: 0.493, corr: 0.203)
- **L14/13393** statistical percentages and survey data (coeff: 0.192, corr: 0.324)
- **L15/25166 themes of neutrality and balance in discourse** (coeff: 0.259, corr: 0.433)
- **L16/21816** phrases related to financial or economic assessments (coeff: 0.543, corr: 0.363)
- L17/5782 references to equality and equity in rights and opportunities (coeff: 0.368, corr: 0.298)
- L18/28196 references to knowledge, learning, and understanding in various contexts (coeff: 0.303, corr: 0.390)
- **L19/29460 discussions about extremes and balance** (coeff: 0.811, corr: 0.440)

- **L20/13319** **expressions of mixed opinions or complex character evaluations** (coeff: 1.413, corr: 0.473)

- L21/8518 references to articles and citations in academic databases (coeff: 2.719, corr: 0.349)

- **L22/28263** **percentages and statistical data concerning opinions or responses** (coeff: 1.024, corr: 0.464)

- **L23/638** formal structures and procedures within organizational contexts (coeff: 1.054, corr: **0.496**)

- **L24/19174** code constructs and control flow keywords related to conditions and returns (coeff: 1.890, corr: 0.465)

- **L25/10753** **expressions of perception or belief in social dynamics** (coeff: 1.147, corr: 0.428)

- L26/27899 code structure and logical operations involving object hierarchy and data types (coeff: 1.025, corr: 0.452)

- **L27/1765** quantitative data related to project development and financial metrics (coeff: 2.597, corr: 0.384)

- L28/21019 financial data and statistics related to development projects (coeff: 0.856, corr: 0.323)

- **L29/17998** code snippets related to JavaScript or Java programming functions and structures (coeff: 1.735, corr: 0.385)

- L30/17084 numerical data related to financial projections and resource development (coeff: 1.308, corr: 0.390)

- L31/10728 auxiliary verbs and words indicating obligation or possibility (coeff: 1.530, corr: 0.239)

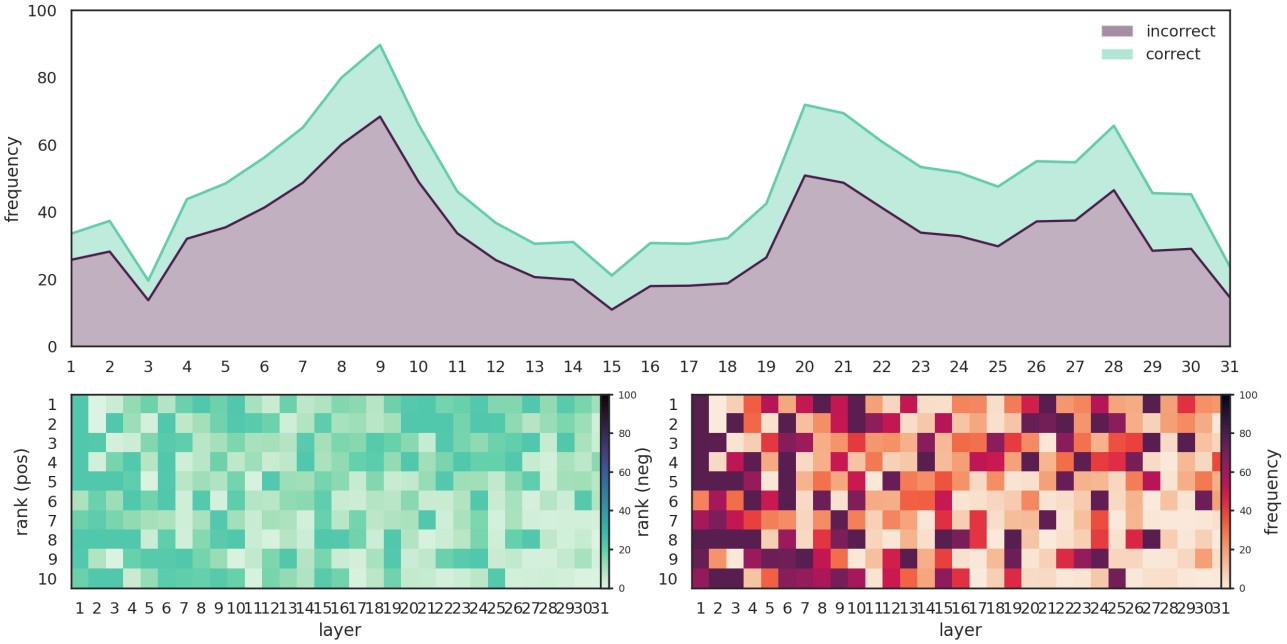

*Figure 23.* Top correlated features with BBQ ambig on frequency in each layer of LLaMA-3.1 8B.

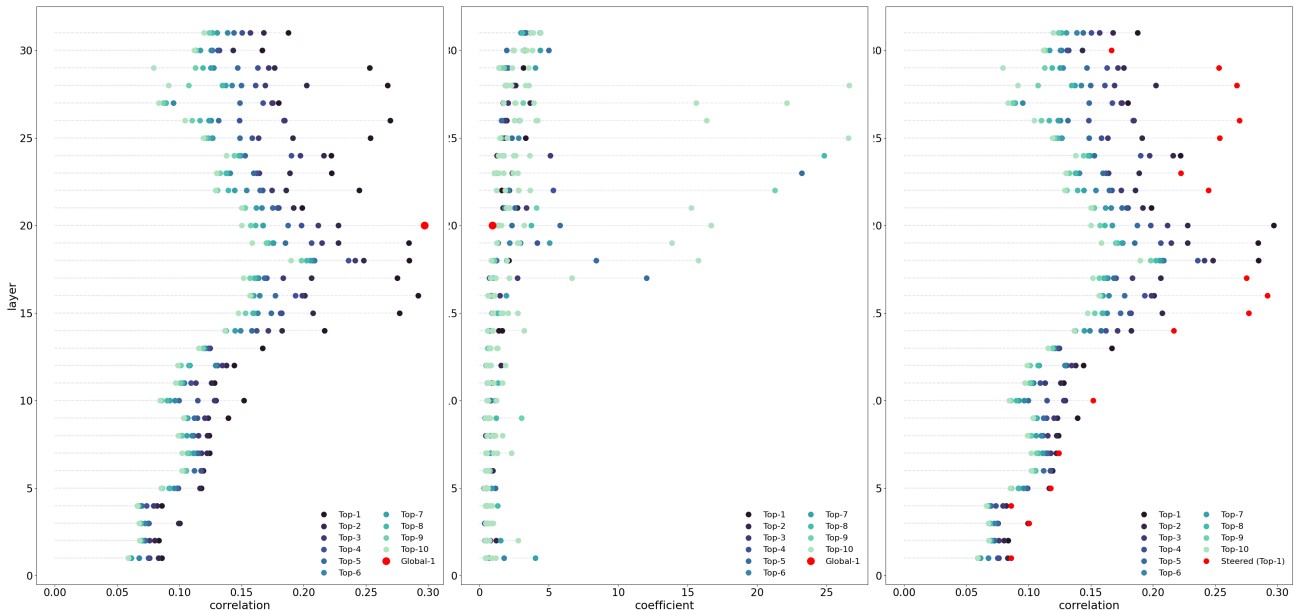

*Figure 24.* Top correlated features with selected features from CorrSteer-P with BBQ disambig on coefficient in each layer of LLaMA-3.1 8B.

## BBQ (Disambiguous)

- **L1/5891** technical terms and references in programming and development contexts (coeff: 0.154, corr: 0.086)
- L2/21865 references to essays, articles, and related writing concepts (coeff: 0.784, corr: 0.084)
- L3/3413 elements related to user engagement and user-friendly design (coeff: 0.332, corr: 0.100)
- **L4/3712** elements related to programming and computation (coeff: 0.458, corr: 0.086)
- **L5/18066** references to educational administration and school district issues (coeff: 0.229, corr: 0.118)
- L6/28294 references to machine learning models and recommendation systems (coeff: 0.301, corr: 0.119)
- **L7/7762** specific language constructs related to coordination and organization (coeff: 0.416, corr: 0.124)
- L8/25466 terms related to hierarchical structures or classifications (coeff: 1.032, corr: 0.124)
- L9/5313 key concepts related to project management and planning (coeff: 0.645, corr: 0.139)
- **L10/13407 negative actions and attitudes that hinder interpersonal relationships and community engagement** (coeff: 0.256, corr: 0.152)
- L11/18350 references to institutions and systems regarding public services (coeff: 0.900, corr: 0.128)
- L12/13336 **phrases and concepts related to community and social interactions** (coeff: 0.377, corr: 0.144)
- L13/15793 negation phrases and words indicating absence or lack (coeff: 0.695, corr: 0.167)
- **L14/31962** details related to physical displacement or movement in a spatial context (coeff: 1.384, corr: 0.217)
- **L15/2128** references to programming elements and constructs (coeff: 0.977, corr: 0.277)
- **L16/6219 code-related syntax and structures within programming languages** (coeff: 0.830, corr: **0.292**)
- **L17/12610** technical terminology related to programming and software development (coeff: 0.706, corr: 0.275)
- L18/16458 HTML tags and structured data elements (coeff: 2.113, corr: 0.285)
- L19/6432 numerical values and the structure of dates or game scores (coeff: 0.909, corr: 0.284)
- **L20/28406** tokens related to timestamps, specifically date and time formats (coeff: 0.942, corr: 0.297)
- L21/15538 references to time management techniques and motivational strategies (coeff: 0.388, corr: 0.199)
- **L22/11286** monetary amounts or financial figures (coeff: 0.531, corr: 0.245)

- L23/30672 phrases involving the concept of answers or responses (coeff: 1.211, corr: 0.222)

- L24/5888 references to answers or responses in discussions or questions (coeff: 1.152, corr: 0.222)

- L25/22713 mathematical notations and symbols (coeff: 1.235, corr: 0.253)

- L26/22133 names of authors and their affiliations in academic contexts (coeff: 1.953, corr: 0.269)

- L27/12321 structural elements and parameters in programming code or data structures (coeff: 0.539, corr: 0.180)

- **L28/23202 specific numbers and their context within factual statements** (coeff: 1.897, corr: 0.267)

- L29/3168 keywords related to health and medical terminology (coeff: 3.175, corr: 0.253)

- L30/22450 terms and phrases related to health and medical conditions (coeff: 3.219, corr: 0.167)

- L31/18173 procedural commands and technical instructions related to software and settings (coeff: 1.440, corr: 0.188)

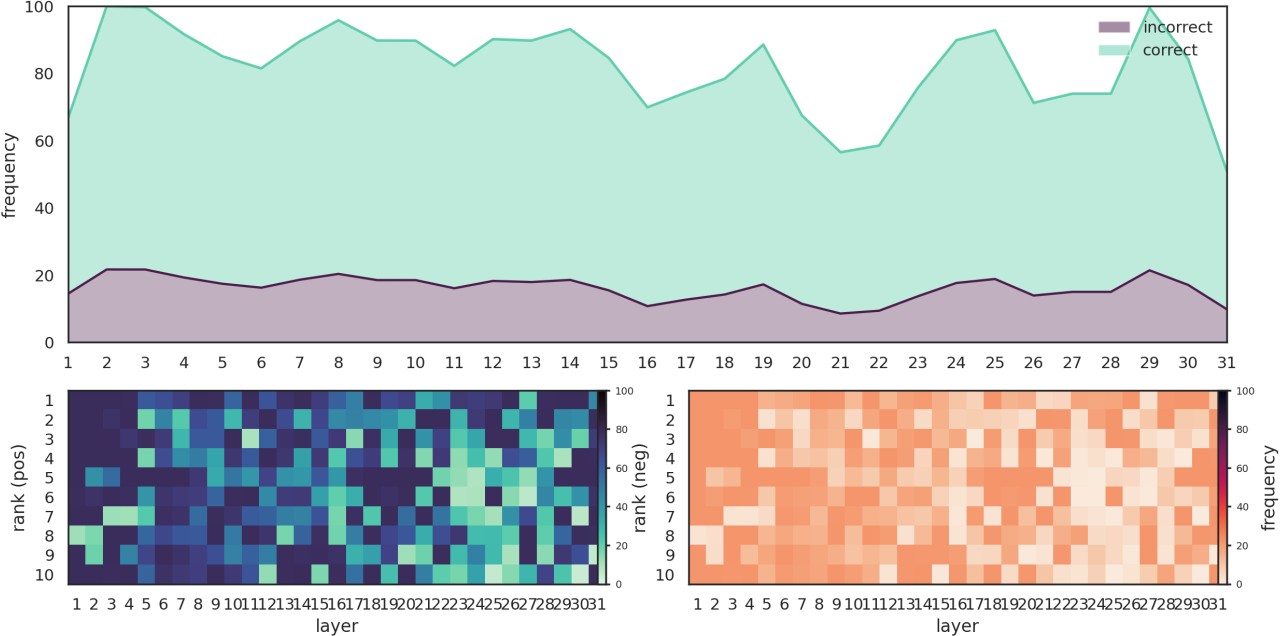

*Figure 25.* Top correlated features with BBQ disambig on frequency in each layer of LLaMA-3.1 8B.

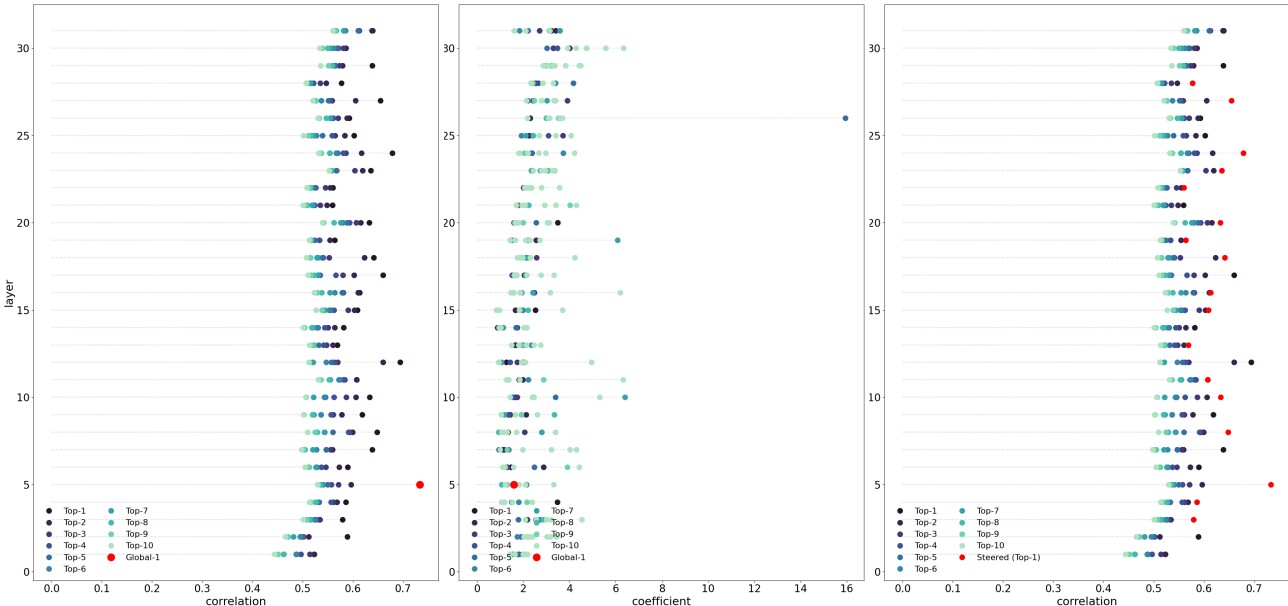

*Figure 26.* Top correlated features with selected features from CorrSteer-P with HarmBench on coefficient in each layer of LLaMA-3.1 8B.

## HarmBench

- `L1/15747` repetitive phrases or expressions related to certainty or emphasis (coeff: 0.491, corr: 0.524)

- `L2/25715` references to collective experiences and communal responsibility (coeff: 1.032, corr: 0.590)

- `L3/23621` **negations and assertions related to existence and actions** (coeff: 1.116, corr: 0.580)

- `L4/26750` **first-person pronouns indicating personal experiences and thoughts** (coeff: 3.468, corr: 0.586)

- `L5/300` **instances of political criticism and hypocrisy** (coeff: 1.587, corr: **0.734**)

- `L6/21616` **discussions about legality, morality, and the implications of actions in ethical contexts** (coeff: 1.458, corr: 0.590)

- `L7/17622` **phrases related to trust and loyalty in political contexts** (coeff: 1.128, corr: 0.639)

- `L8/6508` **expressions related to the condemnation of sexual assault and violence** (coeff: 1.322, corr: 0.648)

- `L9/27026` **concepts related to limits and responsibilities in relationships and societal interactions** (coeff: 1.425, corr: 0.619)

- `L10/9364` **expressions of moral outrage and condemnation regarding social and ethical issues** (coeff: 1.324, corr: 0.633)

- `L11/16561` **expressions of personal opinion and moral judgments** (coeff: 1.810, corr: 0.608)

- `L12/5839` **strong statements against violence and discrimination** (coeff: 1.271, corr: 0.694)

- `L13/15443` emotional expressions of affection or attachment (coeff: 1.637, corr: 0.569)

- `L14/22046` **phrases and sentiments associated with moral judgments and emotional responses** (coeff: 0.750, corr: 0.582)

- `L15/5498` phrases related to environmental and climate impact (coeff: 0.696, corr: 0.609)

- `L16/8375` topics related to stigma and mental health awareness (coeff: 0.938, corr: 0.614)

- `L17/15876` **expressions of self-doubt or uncertainty** (coeff: 0.582, corr: 0.660)

- `L18/6210` phrases related to educational support and challenges faced by teachers (coeff: 0.964, corr: 0.641)

- `L19/5854` references to seeking medical advice and guidance (coeff: 1.148, corr: 0.564)

- `L20/11388` **elements related to moral and ethical dilemmas** (coeff: 3.490, corr: 0.633)

- `L21/9674` **references to racism and social justice issues** (coeff: 0.712, corr: 0.559)

- `L22/4650` expressions of self-awareness and personal growth mixed with skepticism towards collective beliefs (coeff: 2.235, corr: 0.560)

- `L23/28291` **phrases discussing social justice and advocacy for marginalized communities** (coeff: 2.165, corr: 0.636)

- `L24/21055` phrases related to self-identity and personal reflection (coeff: 2.357, corr: 0.679)

- `L25/16450` **themes of emotional struggle and interpersonal relationships** (coeff: 2.415, corr: 0.602)

- `L26/6648` **phrases indicating moral judgment or hypocrisy in political discourse** (coeff: 1.541, corr: 0.593)

- `L27/10654` expressions of emotional conflict and personal reflection (coeff: 1.653, corr: 0.655)

- `L28/522` themes of courage and resilience in writing (coeff: 0.915, corr: 0.578)

- `L29/13883` **complex emotional responses and reflections on interpersonal relationships** (coeff: 2.977, corr: 0.639)

- `L30/4588` **expressions of emotional needs and desires in relationships** (coeff: 1.480, corr: 0.586)

- `L31/31181` **references to familial relationships and memorial details** (coeff: 1.218, corr: 0.639)

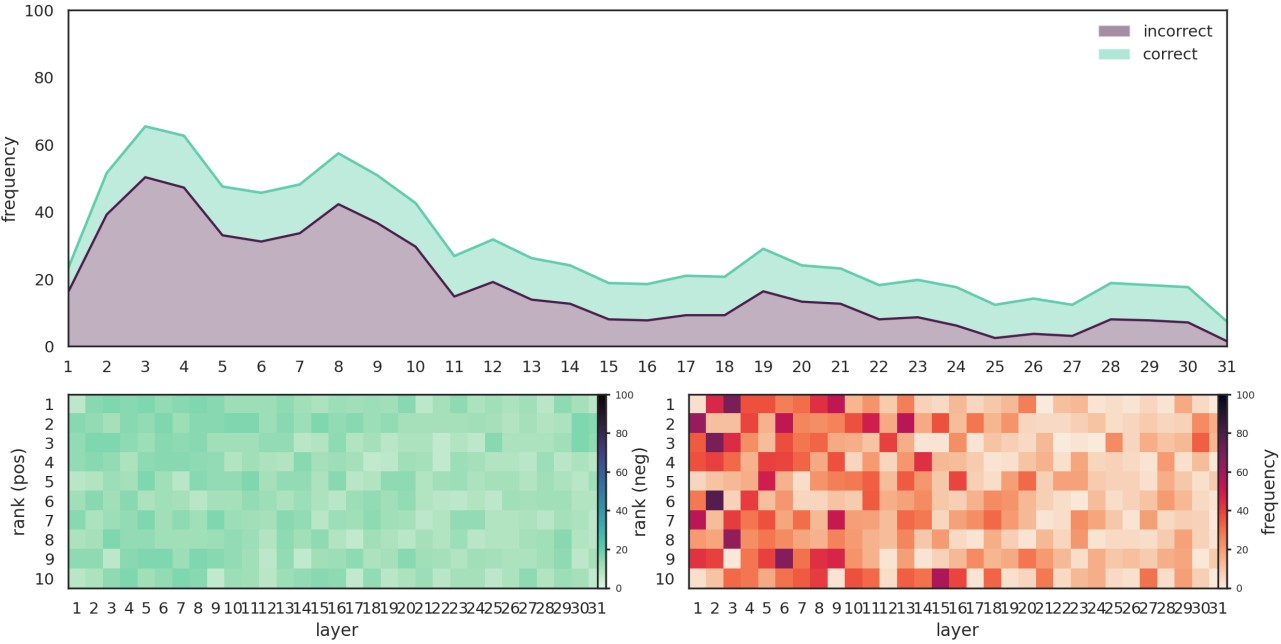

*Figure 27.* Top correlated features with HarmBench on frequency in each layer of LLaMA-3.1 8B.

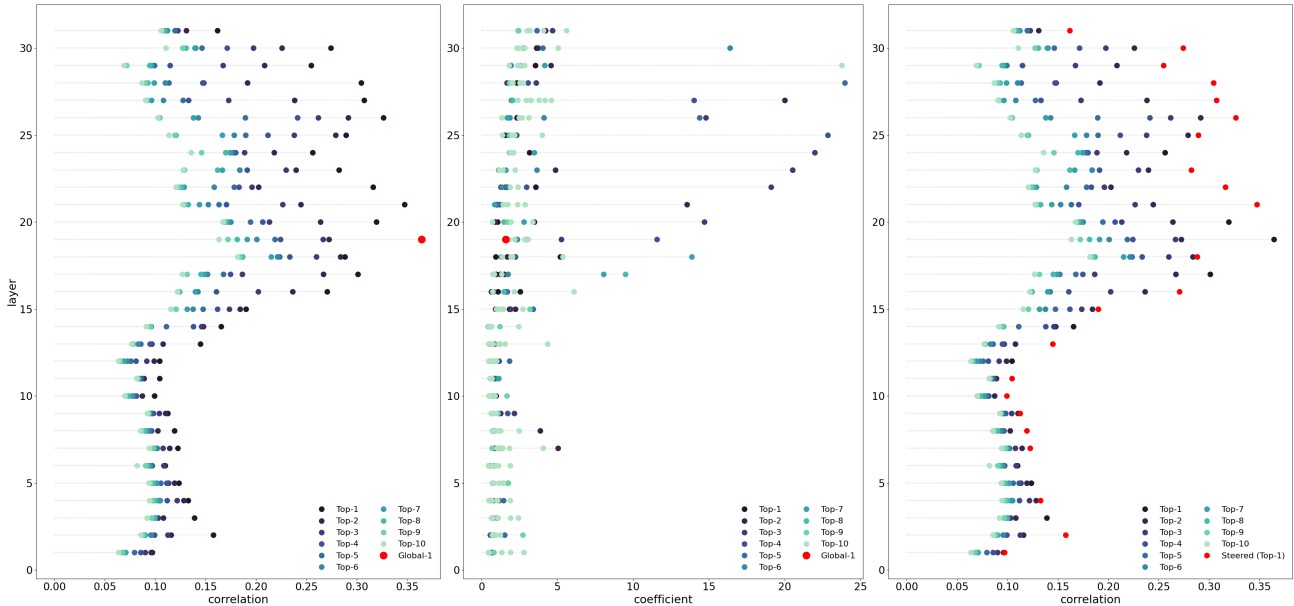

*Figure 28.* Top correlated features with selected features from CorrSteer-P with MMLU ambig on coefficient in each layer of LLaMA-3.1 8B.

## MMLU

- **L1/4557** specific numeric values and measurements related to instructions or guidelines (coeff: 0.695, corr: 0.094)
- **L2/27893** terms related to technology, specifically graphics processing units (GPUs) and their applications (coeff: 0.348, corr: 0.157)
- **L3/204** **terms and concepts related to financial metrics and performance evaluation** (coeff: 1.037, corr: 0.139)
- **L4/23545** questions that lead to detailed inquiries or clarifications (coeff: 1.142, corr: 0.131)
- **L5/17458** **terms related to theoretical concepts and methodologies in scientific discussions** (coeff: 0.497, corr: 0.124)
- **L6/650** specific identifiers, particularly those related to content or lists (coeff: 0.780, corr: 0.110)
- **L7/13659** references to lists, particularly those pertaining to security or classification contexts (coeff: 0.885, corr: 0.118)
- **L8/1649** key terms related to organizational assistance and functionality within various contexts (coeff: 0.871, corr: 0.116)
- **L9/19730** various forms of interviews and discussions related to current events or cultural topics (coeff: 0.397, corr: 0.108)
- **L10/20495** terms related to requirements and definitions within various contexts (coeff: 0.949, corr: 0.099)
- **L11/20851** legal and academic terminology related to charges and reports (coeff: 0.897, corr: 0.100)
- **L12/26346** specific nouns and proper names related to various contexts (coeff: 0.454, corr: 0.104)
- **L13/551** terms related to medical results and actions taken toward health management (coeff: 0.830, corr: 0.143)
- **L14/11013** phrases indicating relationships between people or entities (coeff: 0.366, corr: 0.165)
- **L15/9446** expressions of passion and enthusiasm in various contexts (coeff: 0.327, corr: 0.195)
- **L16/6219** code-related syntax and structures within programming languages (coeff: 1.094, corr: 0.274)
- **L17/26604** references to programming concepts and structures (coeff: 0.957, corr: 0.301)
- **L18/28750** structured data elements and patterns, possibly related to programming or data analysis (coeff: 0.936, corr: 0.288)

- L19/6432 numerical values and the structure of dates or game scores (coeff: 1.587, corr: 0.365)

- L20/28406 tokens related to timestamps, specifically date and time formats (coeff: 1.051, corr: 0.319)

- L21/15538 references to time management techniques and motivational strategies (coeff: 1.014, corr: **0.347**)

- L22/11286 monetary amounts or financial figures (coeff: 1.269, corr: 0.322)

- L23/15096 phrases related to significant life events and milestones (coeff: 1.125, corr: 0.281)

- L24/18010 references to dates and significant life events (coeff: 1.631, corr: 0.256)

- L25/22713 mathematical notations and symbols (coeff: 1.209, corr: 0.287)

- L26/22133 names of authors and their affiliations in academic contexts (coeff: 2.331, corr: 0.331)

- L27/19268 references to academic qualifications, research, and involvement in educational activities (coeff: 0.826, corr: 0.310)

- **L28/23202 specific numbers and their context within factual statements** (coeff: 2.318, corr: 0.307)

- L29/3168 keywords related to health and medical terminology (coeff: 3.545, corr: 0.255)

- L30/23403 terms associated with uncertainty and error (coeff: 0.986, corr: 0.274)

- L31/6722 instances of code-related syntax and formatting (coeff: 0.538, corr: 0.159)

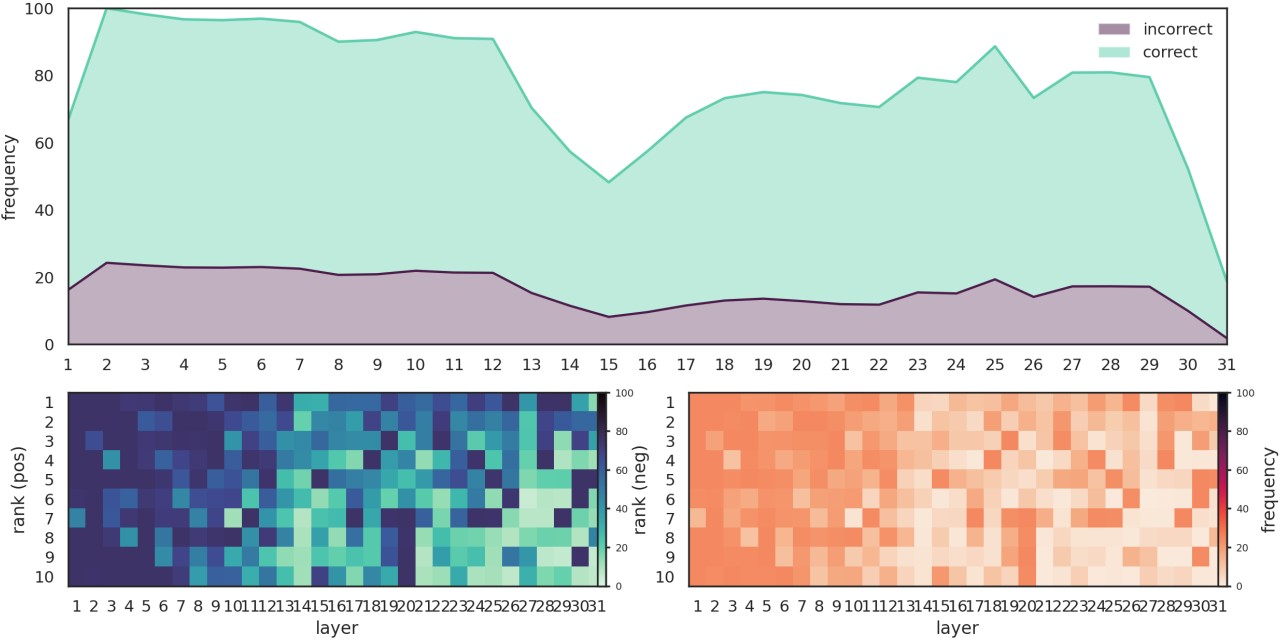

*Figure 29.* Top correlated features with MMLU on frequency in each layer of LLaMA-3.1 8B.

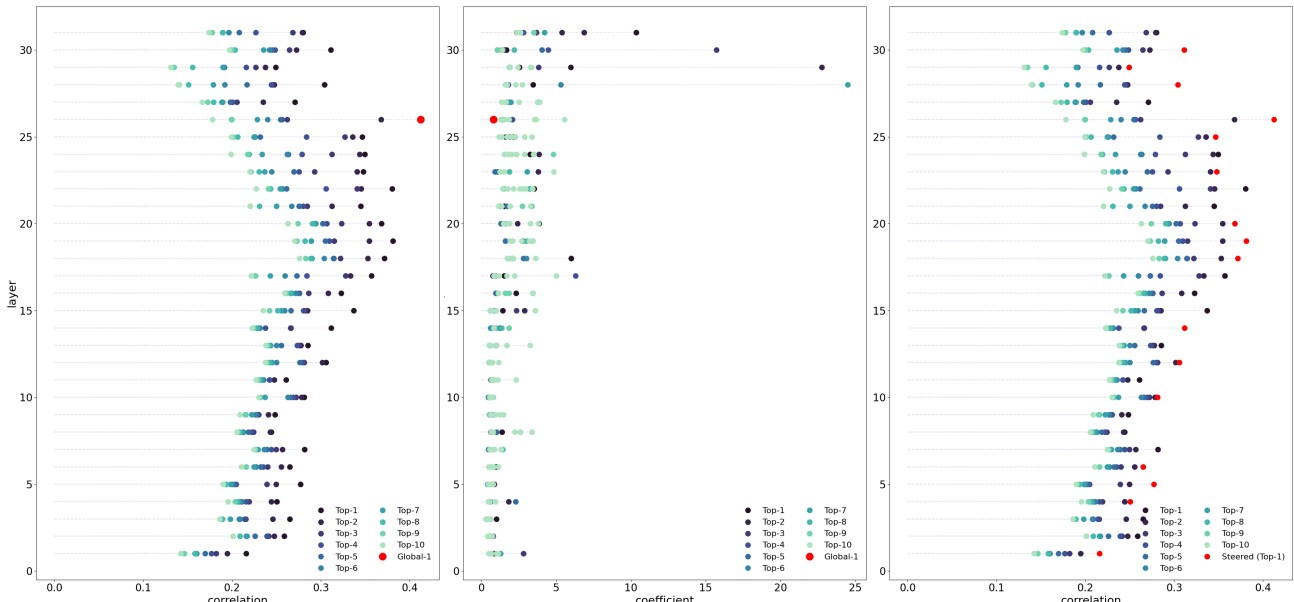

*Figure 30.* Top correlated features with selected features from CorrSteer-P with MMLU-Pro ambig on coefficient in each layer of LLaMA-3.1 8B.

## MMLU-Pro

- L1/2403 specific numeric values and measurements related to instructions or guidelines (coeff: 0.286, corr: 0.216)
- L2/85 phrases related to service expectations and quality assurance
  (coeff: 0.212, corr: 0.259)
- L3/204 terms and concepts related to financial metrics and performance evaluation (coeff: 0.996, corr: 0.265)
- L4/14539 content related to sources and references in articles (coeff: 0.432, corr: 0.250)
- L5/2831 references to urgency and scheduling events (coeff: 0.348, corr: 0.277)
- L6/7784 instances of various relational and transactional terms within context (coeff: 0.153, corr: 0.265)
- L7/22238 references to examples or lists in discussions or reports (coeff: 0.446, corr: 0.282)
- L8/7704 keywords related to television series and their reception
  (coeff: 0.630, corr: 0.244)
- L9/4007 references to various types of businesses and their classifications (coeff: 0.298, corr: 0.248)
- L10/3783 key phrases and concepts related to business development and investment processes (coeff: 0.454, corr: 0.281)
- L11/7301 components of structured data or content organization (coeff: 0.807, corr: 0.261)
- L12/28750 financial terms and conditions related to trading or commerce (coeff: 0.563, corr: 0.306)
- L13/16587 phrases indicating action or involvement in events or developments (coeff: 0.366, corr: 0.285)
- L14/28135 references to specific geographic locations or entities (coeff: 0.490, corr: 0.312)
- L15/9446 expressions of passion and enthusiasm in various contexts (coeff: 0.425, corr: 0.337)
- **L16/6219 code-related syntax and structures within programming languages (coeff: 0.342, corr: 0.323)**
- **L17/26604references to programming concepts and structures (coeff: 0.469, corr: 0.357)**
- L18/2624 references to criminal activity and associated legal consequences (coeff: 0.478, corr: 0.371)
- **L19/6432 numerical values and the structure of dates or game scores (coeff: 0.966, corr: 0.381)**
- L20/28406 tokens related to timestamps, specifically date and time formats (coeff: 0.628, corr: 0.368)

- L21/15538 references to time management techniques and motivational strategies (coeff: 0.391, corr: 0.345)

- L22/11286 monetary amounts or financial figures (coeff: 0.697, corr: 0.380)

- L23/21146 programming and coding structures, particularly related to network protocols and data handling (coeff: 0.853, corr: 0.348)

- L24/7967 references to specific locations or addresses (coeff: 0.837, corr: 0.350)

- L25/16619 instances of authorship and attribution in the text (coeff: 0.864, corr: 0.347)

- **L26/22133 names of authors and their affiliations in academic contexts(coeff: 0.813, corr: 0.413)**

- **L27/19268 references to academic qualifications, research, and involvement in educational activities (coeff: 0.318, corr: 0.271)**

- L28/23202 specific numbers and their context within factual statements (coeff: 1.120, corr: 0.304)

- L29/12442 patterns related to digital platforms and software updates (coeff: 2.528, corr: 0.249)

- L30/19427 specific numerical values and statistical data (coeff: 0.374, corr: 0.311)

- L31/9926 numbers, particularly in relation to financial data and statistics (coeff: 10.348, corr: 0.280)

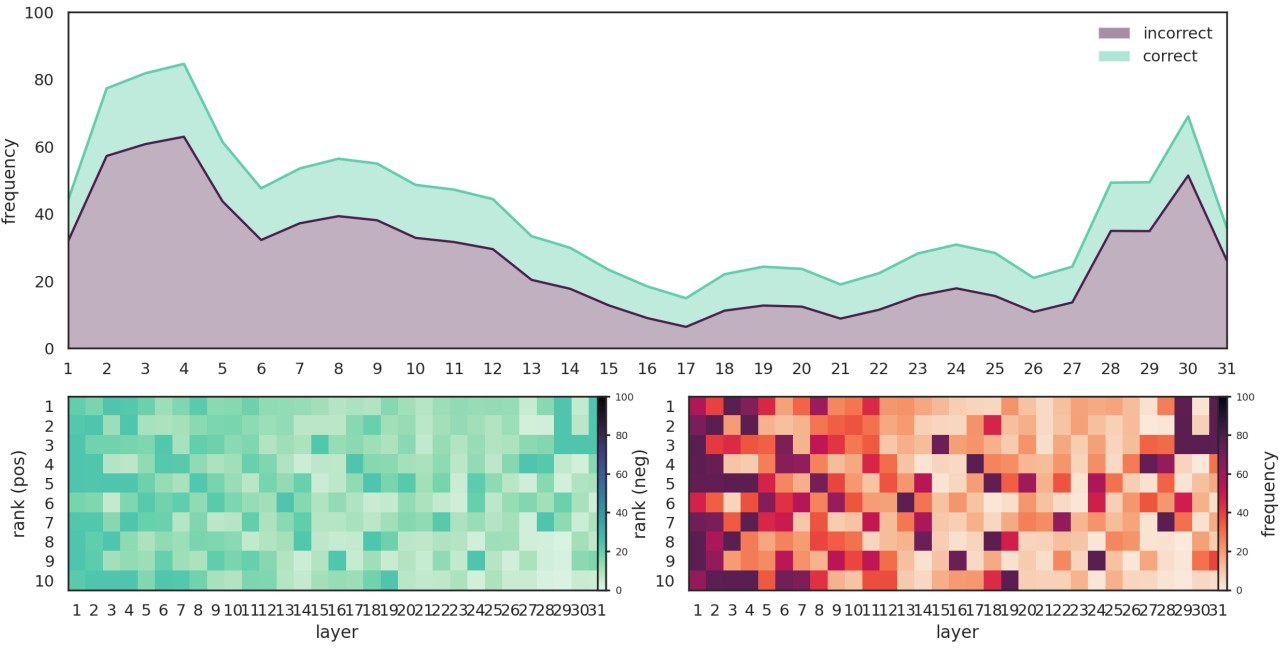

*Figure 31.* Top correlated features with MMLU-Pro on frequency in each layer of LLaMA-3.1 8B.

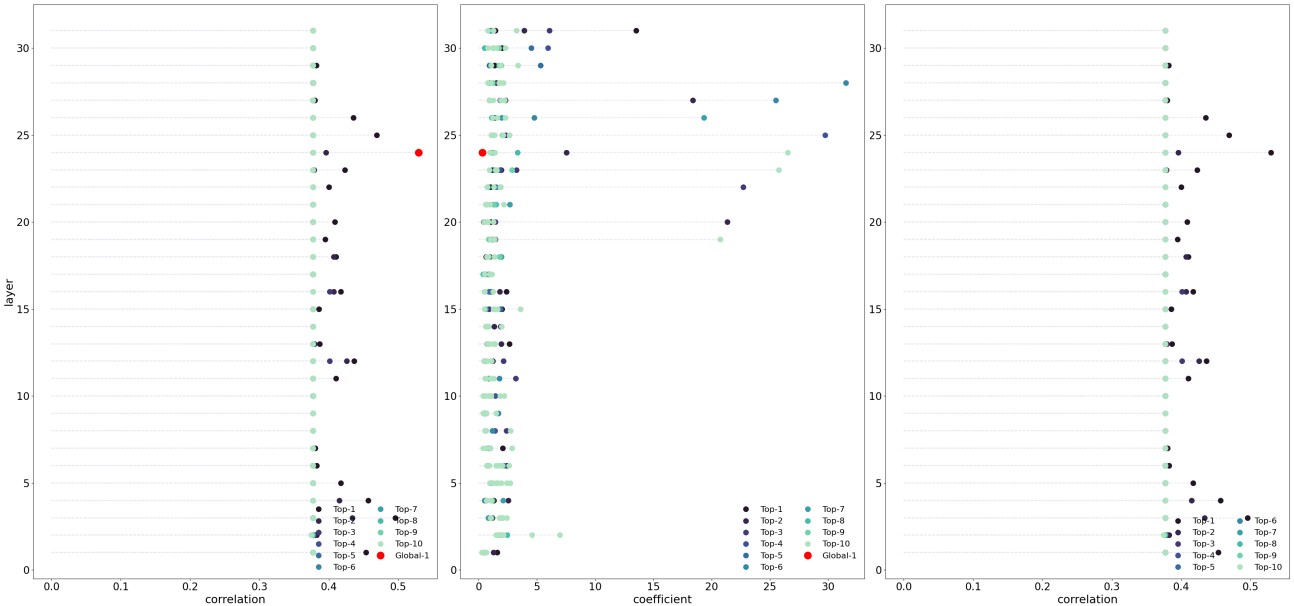

*Figure 32.* Top correlated features with SimpleQA on frequency in each layer of LLaMA-3.1 8B.

## SimpleQA

- `L1/28160` references to height, specifically focusing on the term "tall" (coeff: 1.580, corr: 0.454)
- `L2/16190` references to geographical locations, particularly islands

  (coeff: 0.148, corr: 0.383)
- `L3/24193` references to deserts and desert-related imagery (coeff: 0.541, corr: 0.496)
- `L4/25100` references to dumpster rental services and pricing (coeff: 0.205, corr: 0.457)
- `L5/15924` the occurrence of the word "in" and its context within the text (coeff: 0.396, corr: 0.418)
- `L6/7008` references to artificial entities and technologies (coeff: 2.402, corr: 0.383)
- `L7/6257` terms and phrases related to artificial elements or creations (coeff: 2.049, corr: 0.381)
- `L8/30264` phrases or terms that indicate suitability or excellence in context (coeff: 0.029, corr: 0.377)
- `L9/23784` programming-related keywords and constructs (coeff: 0.089, corr: 0.377)
- `L10/30120` phrases that encourage action or reminders related to specific tasks (coeff: 0.057, corr: 0.377)
- `L11/962` conjunctions that introduce reasoning or causation (coeff: 0.396, corr: 0.410)
- `L12/31391` references to authors and their written works (coeff: 0.472, corr: 0.437)
- **`L13/19013` references to biological family classifications (coeff: 2.618, corr: 0.387)**
- `L14/12579` references to global outreach and international presence (coeff: 0.077, corr: 0.377)
- **`L15/18867` references to biological classifications, specifically family names in taxonomy (coeff: 2.004, corr: 0.386)**
- **`L16/22032` biological classifications of species, particularly family and genus names (coeff: 2.364, corr: 0.417)**
- `L17/30566` phrases related to ownership or affiliation (coeff: 0.884, corr: 0.377)
- `L18/24624` specific terms associated with the media and entertainment industry (coeff: 0.952, corr: 0.410)
- `L19/25841` references to personal growth and transformation experiences (coeff: 1.140, corr: 0.395)
- `L20/23840` references to legislative districts and redistricting processes (coeff: 0.438, corr: 0.409)
- `L21/9851` references to volcanic activity (coeff: 0.258, corr: 0.377)

- L22/20579 references to educational programs and initiatives (coeff: 0.744, corr: 0.400)

- L23/11708 complex arguments and perspectives in academic discourse (coeff: 0.323, corr: 0.423)

- **L24/14877 specific procedural or data-related elements in formal documents (coeff: 0.292, corr: 0.530)**

- L25/18055 words associated with appreciation and commendation (coeff: 0.542, corr: 0.469)

- L26/10617 emotional expressions and relationships in personal narratives (coeff: 0.317, corr: 0.435)

- L27/135 activities related to travel and tourism (coeff: 0.924, corr: 0.380)

- L28/29877 references to the concept of "home." (coeff: 0.964, corr: 0.377)

- L29/4392 references to clothing and dress codes, particularly in relation to gender identity and expression (coeff: 0.410, corr: 0.382)

- L30/22633 public methods in a programming context (coeff: 0.310, corr: 0.377)

- L31/6171 references to artificial intelligence and its related concepts (coeff: 1.429, corr: 0.377)

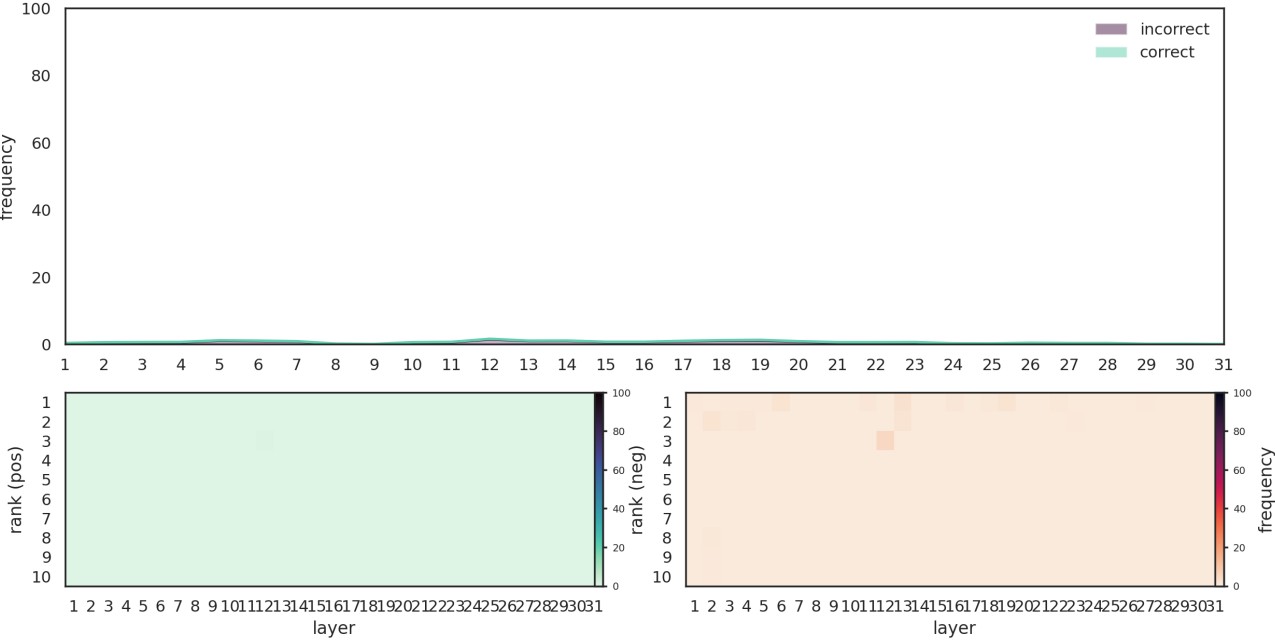

*Figure 33.* Top correlated features with SimpleQA on frequency in each layer of LLaMA-3.1 8B.

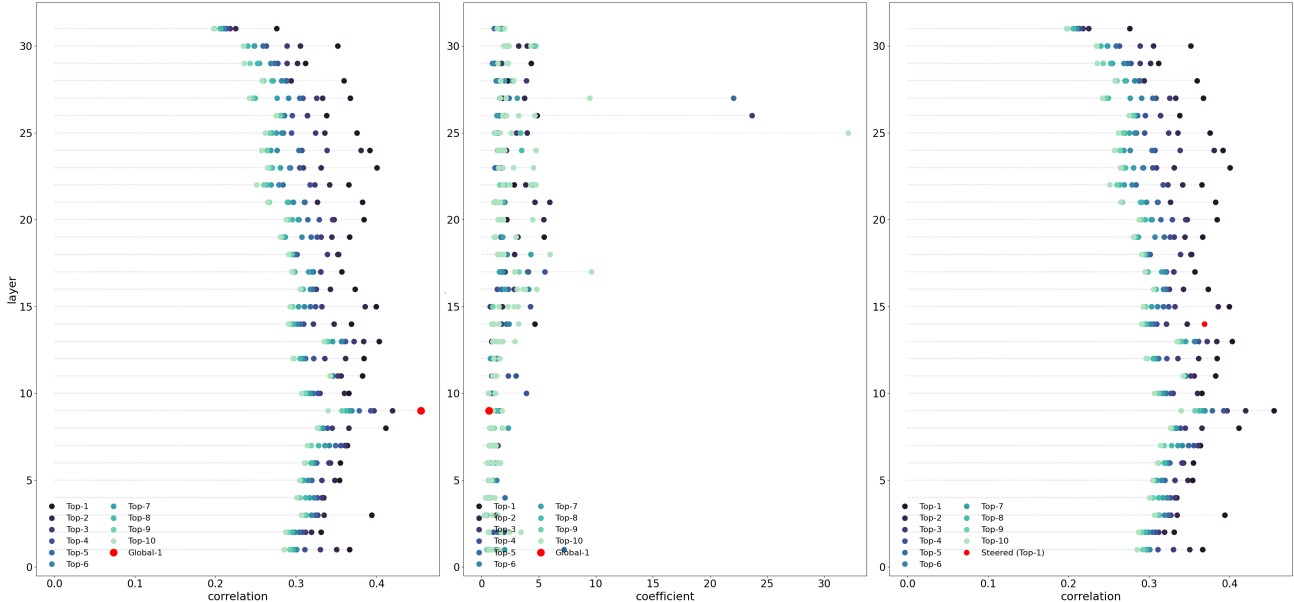

*Figure 34.* Top correlated features with XSTest on frequency in each layer of LLaMA-3.1 8B.

## XSTest

- L1/6754 references to studies and publications (coeff: 0.256, corr: 0.367)

- L2/5332 names and characteristics associated with aviation or flight (coeff: 0.276, corr: 0.331)

- L3/16461 terms related to marine life and conservation efforts (coeff: 1.265, corr: 0.394)

- L4/2446 proper nouns and specific entities (coeff: 0.310, corr: 0.334)

- L5/25000 names of notable individuals and places related to historical or cultural significance (coeff: 0.862, corr: 0.354)

- L6/10424 information related to personal details and statistics about individuals (coeff: 0.220, corr: 0.355)

- L7/20235 words and phrases associated with measurement or assessment (coeff: 0.784, corr: 0.364)

- L8/22807 concepts related to capital budgeting and investment decision-making (coeff: 0.420, corr: 0.411)

- **L9/16423 references to specific organizations, laws, or conditions related to societal issues (coeff: 0.636, corr: 0.455)**

- L10/11238 phrases related to collaboration and community involvement (coeff: 0.880, corr: 0.365)

- **L11/29172 legal terminology related to civil rights and obligations (coeff: 0.618, corr: 0.383)**

- **L12/19663 negative descriptors or concepts related to cowardice and existence (coeff: 0.735, corr: 0.384)**

- L13/19506 numeric or alphanumeric strings and specific identifiers (coeff: 0.608, corr: 0.403)

- **L14/13505 structured question-answer formats and indicators of a discussion or inquiry (coeff: 4.659, corr: 0.369)**

- **L15/23853 references to female characters and their relationships in narratives (coeff: 0.682, corr: 0.400)**

- L16/1652 names and identifiers related to locations and organizations (coeff: 1.220, corr: 0.373)

- L17/21476 references to influential figures in scientific history and significant concepts from their work (coeff: 2.046, corr: 0.357)

- L18/25543 names and specific references related to individuals, locations, and organizations in a political context (coeff: 0.941, corr: 0.353)

- L19/2102 significant historical events and their impact on society (coeff: 1.691, corr: 0.366)

- L20/21486 various references to awards, accolades, and notable achievements within literary and cinematic contexts (coeff: 2.183, corr: 0.385)
- L21/8477 references to influential figures and their contributions in various contexts (coeff: 2.008, corr: 0.383)
- L22/16870 references to disasters and their impacts (coeff: 2.837, corr: 0.366)
- L23/15524 references to specific events or characters in films (coeff: 1.834, corr: 0.400)
- L24/15231 references to specific events or characters in films (coeff: 1.747, corr: 0.392)
- L25/16855 references to corporate entities and financial transactions (coeff: 0.763, corr: 0.375)
- L26/1578 references to specific individuals or organizations involved in social causes or environmental conservation (coeff: 0.948, corr: 0.338)
- L27/11758 connections to authoritative figures and organizational roles (coeff: 1.300, corr: 0.367)
- L28/425 instances of specific names and organizational references in a text (coeff: 2.291, corr: 0.360)
- L29/17372 terms related to health and illness (coeff: 0.888, corr: 0.312)
- L30/11223 titles and descriptors of programs or services related to community support (coeff: 4.643, corr: 0.352)
- L31/2111 descriptions and features of software products (coeff: 1.614, corr: 0.276)

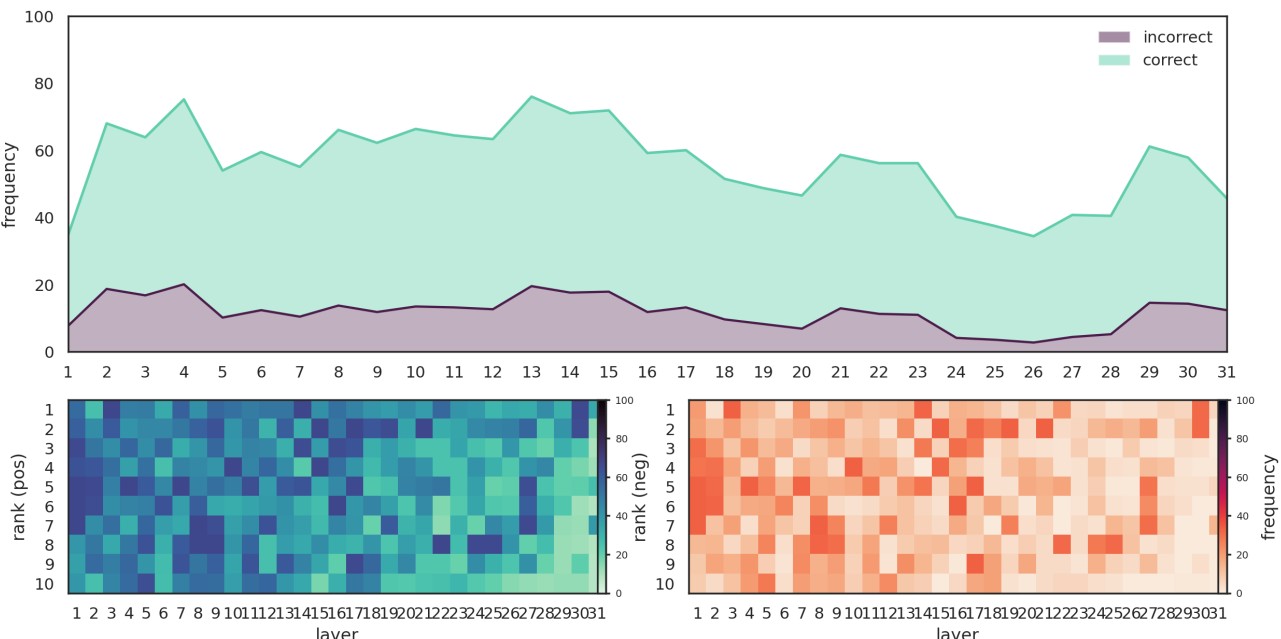

*Figure 35.* Top correlated features with XSTest on frequency in each layer of LLaMA-3.1 8B.

